



# A first predictive mechanistic model of cold-water coral biomass and respiration based on physiology, hydrodynamics, and organic matter transport

Evert de Froe[1, 2, 3*], Christian Mohn[4], Karline Soetaert[5], Anna-Selma van der Kaaden[6], Gert-Jan Reichart[1,7], Laurence H. De Clippele[8], Sandra R. Maier[9], Dick van Oevelen[5]

[1]Department of Ocean Systems, NIOZ Royal Netherlands Institute for Sea Research, Texel, the Netherlands
[2] Centre for Fisheries Ecosystem Research, Fisheries and Marine Institute at Memorial University of Newfoundland, St. John's, NL, Canada
[3]Wageningen Marine Research, Yerseke, the Netherlands
[4]Department of Ecoscience, Aarhus University, Roskilde, Denmark
[5]Department of Estuarine and Delta Systems, NIOZ Royal Netherlands Institute for Sea Research Yerseke, the Netherlands
[6] Copernicus Institute of Sustainable Development, Faculty of Geosciences, Utrecht University, Utrecht, The Netherlands
[7] Utrecht University, Faculty of Geosciences, Utrecht, the Netherlands
[8] School of Biodiversity, One Health & Veterinary Medicine, University of Glasgow, Glasgow, United Kingdom
[9] Greenland Climate Research Centre, Greenland Institute of Natural Resources, Nuuk, Greenland

*Corresponding author: evert.defroe@wur.nl

**Abstract.** Cold-water corals form complex three-dimensional structures on the seafloor, providing habitat for numerous species and act as a carbon cycling hotspot in the deep-sea. The distribution of those important ecosystems is often predicted by statistical habitat suitability models, using variables such as terrain characteristics, temperature, salinity, and surface productivity. While useful, these models do not provide a mechanistic understanding of the processes that facilitate cold-water coral occurrence, and how this may change in the future. Here, we present the results of a mechanistic process-based model in which coral biomass and respiration are predicted from a 3D coupled transport-reaction-model for south-east Rockall Bank (NE Atlantic Ocean). Hydrodynamic forcing is provided by a high-resolution Regional Ocean Modelling System (ROMS) model, which drives the transport of reactive suspended particulate organic matter in the region. The physiological cold-water coral model, with coral food uptake, assimilation, and respiration as key variables and with model parameters estimated from available experimental report, is coupled to the reactive transport model of suspended particulate organic matter. Model predictions agree with coral reef biomass and respiration observations in the study area and coral occurrences comply with predictions from previously published habitat suitability models. Cold-water coral biomass was mainly predicted on coral mounds and ridges in the area. Filter feeding activity by cold-water corals proved to strongly deplete food particles in the bottom waters. Replenishment of food particles by tidal currents was therefore vital for cold-water coral growth. This mechanistic modelling approach has the advantage over statistical and machine learning-based predictions that it can be used to obtain an understanding of the effect of changing environmental conditions such as ocean





temperature, surface production export, or ocean currents on cold-water coral biomass distribution and can be applied to other study areas and/or species.

**Key findings**

- First mechanistic model predicting cold-water coral biomass distribution based on both organic matter transport and hydrodynamics
   - High cold-water coral biomass occurred in areas with sufficient organic matter replenishment and high bottom current speed.
   - Benthic carbon mineralization rates on the coral mounds comprised mostly of cold-water coral respiration and
compare well with observations.
   - Coupling organic matter uptake with the cold-water coral model was key in predicting a realistic spatial distribution of cold-water corals.

**Short summary**

Cold-water corals are important reef-building animals in the deep sea, and are found all over the world. So far, researchers
have been mapping and predicting where cold-water corals can be found using video transects and statistics. This study provides the first process-based model in which corals are predicted based on ocean currents and food particle movement. The renewal of food by tidal currents close to the seafloor and corals proved essential in predicting where they can grow or not.

## 1   Introduction

Scleractinian cold-water corals (CWCs) are ecosystem engineers in the deep-sea that build reefs of high biodiversity and biomass (Greiffenhagen et al., 2025; Henry and Roberts, 2007; Jonsson et al., 2004; Roberts et al., 2006). These reefs are considered hotspots of organic matter remineralization in the deep-sea (Cathalot et al., 2015; de Froe et al., 2019; Rovelli et al., 2015), and can form carbonate mounds over geological timescales (Dorschel et al., 2005; van der Land et al., 2014; Wienberg et al., 2020). CWC reefs and carbonate mounds are distributed globally (Davies and Guinotte, 2011) in a wide
range of environmental conditions, e.g., temperature and oxygen (Dullo et al., 2008; Flögel et al., 2014; Hanz et al., 2019; Mienis et al., 2012), but typically in areas with high bottom currents (Davies et al., 2009; Genin et al., 1986; Juva et al., 2020; Mienis et al., 2007). Thriving CWC reefs have been linked to relatively low dissolved inorganic carbon (DIC) concentrations, i.e., a high carbonate saturation state (Flögel et al., 2014), strong tidal currents (Juva et al., 2020), and an above-average quantity (Guinotte et al., 2006) and quality supply of food (de Froe et al., 2022; Kiriakoulakis et al., 2007).
CWC reefs, and deep-sea habitats in general, are currently under threat by climate-induced oceanic change (Brito-Morales et al., 2020; Gehlen et al., 2014; Jones et al., 2014; Levin and Le Bris, 2015; Morato et al., 2020), such as ocean acidification



(Caldeira and Wickett, 2003; Hennige et al., 2015, 2020), increasing temperatures (Gómez et al., 2022; Maier et al., 2023; Wijffels et al., 2016), altered large scale ocean circulations (Boers, 2021; Caesar et al., 2021), and decreased carbon export from the surface layer (Bopp et al., 2001; Laws et al., 2000; Wohlers et al., 2009). However, predicting how these

environmental changes will affect deep-sea habitats remains challenging due to, for instance, technical constraints in sampling possibilities, and new tools and models are needed to understand how CWCs will be influenced by a changing marine environment.

Current capacity to predict the spatial distribution of CWCs is limited to statistical and machine learning approaches, which

can predict the probability of CWC occurrence, density and biomass based on a broad set of environmental variables (i.e., terrain variables, depth, pH, temperature, salinity; e.g., De Clippele et al., 2021b, a; Greiffenhagen et al., 2024; Guinotte and Davies, 2014; Morato et al., 2020; Rengstorf et al., 2014). These are supervised learning approaches, where a model is provided with observations of CWC and accompanying environmental variables, from which their probability to areas without observations are predicted. While these statistical models can be used to understand the consequences of climate

change for CWCS (i.e., Morato et al., 2020), they offer limited mechanistic understanding of underlying processes that drive spatial distribution of species under climate change outcomes (Evans et al., 2015).

A mechanistic modelling approach, in which interactions are explicitly described by process-based formulations, can help understand what processes are important in predicting the spatial distribution of CWC species. Organic matter uptake by

CWC reefs alter the food flux towards the reefs, thereby affecting their own growth and spatial distribution (van der Kaaden et al., 2020, 2024; Soetaert et al., 2016a). The organic matter availability is enhanced on the reefs and depleted around the reefs. This feedback between organisms and their environment can greatly affect how they respond to environmental changes: by modifying their own environment, organisms can rearrange their spatial patterns in response to climate change thereby avoiding a tipping point towards extinction (Rietkerk et al., 2021). However, with CWC reefs, their response to

climate change is typically predicted with statistical models that extrapolate the predicted distribution of suitable habitat into the future. Such methods do not take into account the organism-environment interaction that has been shown to be key to understanding an organism's response to climate change.

The development of a mechanistic model requires sufficient knowledge of the ecosystem functioning and of the relevant

species' physiology and food supply mechanisms. Mechanistic modelling efforts so far have been able to model hydrodynamics at CWC mounds (van der Kaaden et al., 2021; Mohn et al., 2014), as well as organic matter transport in the water column (Soetaert et al., 2016a). Recently, surface productivity, food supply, and local hydrodynamics have been identified as the most important factors for coral growth (De Clippele et al., 2021a, b; Fink et al., 2013; Hebbeln et al., 2019; Maier et al., 2023) within their environmental niche (determined by e.g., temperature, oxygen; e.g., Dullo et al., 2008; Hanz

et al., 2019). CWC mounds alter local hydrodynamics (Cyr et al., 2016; van der Kaaden et al., 2021), thereby enhancing



water column mixing and consequently food supply toward the mounds (van der Kaaden et al., 2020; Soetaert et al., 2016a). Filtering activity of CWC reefs also depletes Particulate Organic Carbon (POC) in surrounding bottom waters (Lavaleye et al., 2009; Wagner et al., 2011), which may lead to spatial self-organization of CWC reefs and mounds (van der Kaaden et al., 2020, 2023). Our understanding of CWC reefs and CWC physiology has vastly improved in the past decades i.e., their basal-
and total respiration rates (Dodds et al., 2007; Larsson et al., 2013; Maier et al., 2019), food uptake rates (Gori et al., 2014), feeding behaviour under altering current speeds (Orejas et al., 2016a), and *in situ* polyp behaviour in relation to tidal currents (Osterloff et al., 2019). Here we use these insights to build on an earlier developed modelling framework that simulated local hydrodynamic forcing and organic matter transport (Mohn et al., 2014, 2023; Soetaert et al., 2016a), to predict the spatial distribution of CWC biomass and benthic respiration in a CWC mound region in the northeast Atlantic Ocean.


The CWC mounds and ridges on the south-eastern (SE) slope of Rockall Bank (northeast Atlantic Ocean) provide an excellent study site to develop a mechanistic model of CWC biomass and respiration in relation to local hydrodynamics and food supply. This area has strong regional contrasts with numerous CWC mounds between 500-1000 m depth surrounded by sediments, which have been studied extensively for several decades (de Froe et al., 2022; van Haren et al., 2014; Kenyon et
al., 2003; Mienis et al., 2006). The mounds are formed by the framework building CWC species *Desmophyllum pertusum* (previously known as *Lophelia pertusa*, Addamo et al., 2016) and *Madrepora oculata*, for which a relatively large amount of physiological data is available. Past mechanistic modelling efforts at Rockall Bank include a local high-resolution setup of the 3D hydrodynamic Regional Ocean Modelling System (ROMS – AGRIF; Shchepetkin and McWilliams, 2005; Mohn et al., 2014) and an organic matter transport model (Soetaert et al., 2016a). A machine learning classification-based biomass
modelling approach predicted CWC biomass, carbon stock, and nutrient cycling capacity of the area (De Clippele et al., 2021a, b). More recently, new 3D hydrodynamic ROMS output has been developed to investigate changes of the Atlantic Meridional Overturning Circulation (AMOC; Mohn et al., 2023), and to study how coral mound size affects local hydrodynamics (van der Kaaden et al., 2021, 2024). Furthermore, recent observational studies on benthic carbon cycling (de Froe et al., 2019), video transects (De Clippele et al., 2019; Maier et al., 2021), hydrodynamics (Schulz et al., 2020), and a
CWC habitat suitability model of the Logachev mound area (Rengstorf et al., 2014) offer the opportunity to validate our mechanistic modelling predictions.

Here, we present a coupled mechanistic model based on hydrodynamics, organic matter transport, and CWC physiology. We specifically aim to predict CWC biomass and quantify CWC and benthic respiration. Moreover, we investigate what drives
the spatial distribution of CWCs by examining the influence of bottom current speed, organic matter transport, and organic matter depletion on the distribution of CWC biomass in the study area. Our study contributes to the development of a modelling approach which can be used to investigate how CWCs will be affected by changing oceanic conditions. Below, we first describe the modelling approach and data sources that were used to validate the model output. Then, we show how our model performs compared to observational data on 1) hydrodynamics, 2) POC concentration and transport, 3) CWC





spatial distribution, and 4) benthic respiration. Finally, we discuss the spatial distribution of predicted CWC biomass in the model domain, compare our model predictions with machine learning modelling methods, and discuss the limitations and prospects of our model results.

## 2    Material and Methods

### 2.1    Study area

Our study area is situated on the south-eastern (SE) slope of Rockall Bank (north-east Atlantic; Figure 1A). The substrate in this area is characterized by biogenic soft sediment at the shallow part of Rockall Bank (300 – 500 m depth), coral capped carbonate mounds and ridges on the slope between 500 – 1000 m depth, and biogenic soft sediments in between the carbonate mounds and in the deeper part of the Rockall Bank slope (>1000 m depth; Kenyon et al., 2003; Mienis et al., 2006). The current direction throughout the water column is predominantly to the southwest, driven by the clockwise gyre

circling the Rockall Bank (Hansen and Østerhus, 2000; Holliday et al., 2000; Mienis et al., 2007; Schulz et al., 2020). The area is subject to internal waves with amplitude of several 100s of meters and high bottom current speeds (i.e., >50 cm s$^{-1}$; Mienis et al., 2007; Mohn et al., 2014). Interaction of tidal currents with mound topography cause breaking of internal waves (Cyr et al., 2016) with subsequent downward transport of organic matter (Duineveld et al., 2007; de Froe et al., 2022; Soetaert et al., 2016a). In our study area, coral ridges are found in the north-eastern part, while numerous coral mounds,

known as the Logachev mound province, are found in the southwest (Figure 1B). For readability, we will refer to the coral mounds and ridges in this area as 'CWC mounds'. The largest CWC mound in the model domain is called "Haas mound" which is around 500 m high, one to two km wide, five km long, and elongated parallel to the Rockall Bank slope.





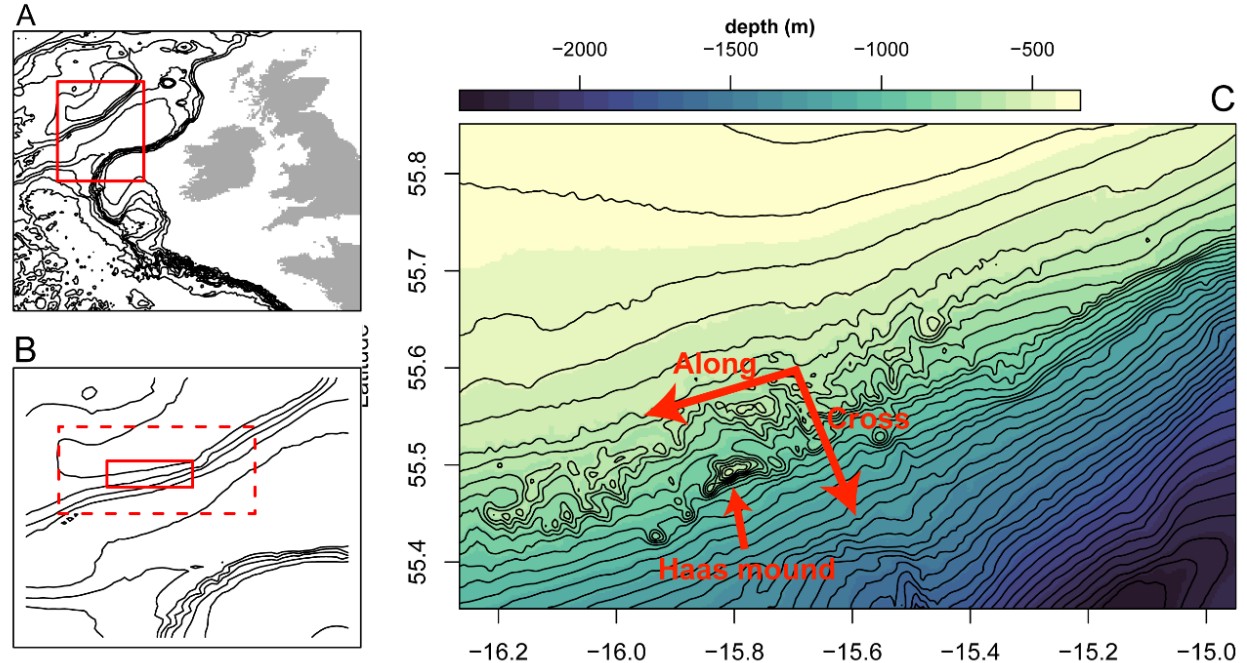

**Figure 1: Study area and model domain. (A) location study area in the Atlantic Ocean, the red line marks the plot boundaries of**
**panel B, (B) location of parent grid (dashed lines) and the embedded child grid (solid line), (C) model domain with dimensions of**
**83 x 55 km, and 450 – 2300 m depth range. The red arrows indicate the definition of cross- and along slope current directions on**
**Rockall Bank (as also used in Schulz et al., 2020).**

## 2.2    Model species: *Desmophyllum pertusum*

The carbonate ridges and mounds in the study area are formed by the framework building CWCs *Desmophyllum pertusum*
and *Madrepora oculata* (Duineveld et al., 2007; Maier et al., 2021; van Weering et al., 2003). These corals form bush-like
colonies that can grow several metres high consisting of thousands of coral polyps (Roberts et al., 2009). In this study, *D.
pertusum* is used as model species as it is considered a keystone species, providing habitat to numerous associated animals
(Costello et al., 2005; Freiwald et al., 2002; Husebø et al., 2002; Jensen and Frederiksen, 1992), and it contributes
substantially to reef metabolism (de Froe et al., 2019). *Desmophyllum pertusum* generally feeds on a mixture of particulate
organic matter and zooplankton, and in the Rockall Bank area primarily on particulate organic matter (Duineveld et al.,
2007), as local zooplankton density is relatively low (de Froe et al., 2022). Therefore, we here use suspended particulate
organic carbon (POC) as food source proxy for *D. pertusum*.

## 2.3    Modelling approach

This study was set-up by coupling three models (details provided below): first, hydrodynamic output was extracted from a
local setup of the ROMS-AGRIF model, which was previously developed and validated for the Rockall Bank study region
(Mohn et al., 2014, 2023). Secondly, the hydrodynamics forced a suspended particulate organic carbon (POC) reactive-




transport model following Soetaert et al. (2016). Finally, a new physiologically-based CWC and sediment model was developed, using the POC dynamics and hydrodynamic model data as forcing, to predict CWC biomass, sediment organic matter content, and CWC- and sediment respiration.

### 2.3.1 Hydrodynamic model

Mohn et al. (2014) gives a detailed description of the hydrodynamical modelling set-up of which the output was used in this study. In short, the ROMS-AGRIF model domain encompasses an area of 86 x 58 km, which was divided into 336 grid cells in longitudinal direction and 228 grid cells in meridional direction, resulting in a horizontal resolution of 250 meters (depth range 320-2500 meters; Figure 1B). This child model grid was nested inside a parent grid with a 750 m resolution that encompassed an area of 190 x 188 km (Figure 1B). Both models were run on a staggered Arakawa-C grid with a terrain-following stretched sigma grid of 32 vertical levels (Supplementary Figure A1) which provides a higher vertical resolution at the surface and seafloor (Shchepetkin and McWilliams, 2005). Bottom layer height was on average around 8 m. Tidal forcing was prescribed using the TPXO7 global tidal inverse solution (Egbert and Erofeeva, 2002). Here we used hydrodynamic output (horizontal currents) data for the period March 1979, which was simulated and validated in a separate study investigating the influence of contrasting AMOC states on benthic hydrodynamics at different CWC sites (Mohn et al., 2023). This specific time period was chosen for two purposes: first, fresh POC arrives at CWC reefs in the study area in spring (Duineveld et al., 2007). Second, in the year 1979 the AMOC showed a weak signal (Böning et al., 2016). Given that the AMOC state is currently at its weakest of the last millennium (Caesar et al., 2021), and may weaken further under global carbon emission projections (Bakker et al., 2016; Caesar et al., 2018), using modelled properties of a weak AMOC year provides a good representation of future large scale circulation patterns and hydrodynamic conditions (Mohn et al., 2023). The hydrodynamic output, stored at 3-hour intervals, was then used to model POC dynamics in the water column.

### 2.3.2 Organic matter transport model

A detailed description of the organic matter transport model can be found in Soetaert et al. (2016). In this model, we predict the transport of particulate organic carbon (POC) through the water column and deposition on the seafloor. POC is transported through the model domain by a combination of advective and passive transport. POC advection occurs throughout the model domain based on the 3-hour interval output from the ROMS-AGRIF model, with zero-gradient boundary conditions on the lateral boundaries of the model domain. A constant input of 12 mmol POC $m^{-2}$ $d^{-1}$ is assumed at the upper boundary (i.e., export from the photic zone)/ This value is based on the annual primary production and export efficiency in the study area and we used an annual primary production and export efficiency for the Rockall Bank area of 200 g C $m^{-2}$ and 25 % respectively (Henson et al., 2015), resulting in an export of 12 mmol C $m^{-2}$ $d^{-1}$. Passive transport occurs by POC passive sinking of 10 m $d^{-1}$ through the water column, which is considered representable for slow sinking suspended particles (Riley et al., 2012). Deposition of POC on the seafloor is assumed as bottom POC concentration times the passive sinking rate. Furthermore, POC is subject to constant first-order decay. The degradation rate of sinking POC is



defined as $k = 0.016 \cdot 1.066^T$ (Henson et al., 2015; Yool et al., 2011), in which T is temperature in °C. Temperature ranges

from 4 °C at 2000 m depth to 15 °C in the summer periods at the surface. Therefore, we used a water column temperature of

10°C which corresponds to a $k$ of 0.03 d$^{-1}$. The reactive-transport model is implemented as:

$$\frac{dH_z \cdot POC}{dt} = -\frac{d(H_z \cdot u \cdot POC)}{dx} - \frac{d(H_z \cdot v \cdot POC)}{dy} - \frac{d(H_z \cdot w \cdot POC)}{dz} - \frac{d(H_z \cdot w_s \cdot POC)}{dz} -$$

$$k \cdot H_z \cdot POC \text{ (Equation 1)}$$

Where H$_z$ is the grid cell thickness in meters, POCC is the concentration POC in the grid cell in mmol C m$^{-3}$, $u$ is the

eastward horizontal water velocity, $v$ is the northward horizontal water velocity, $w$ is the vertical water velocity, $w_s$ is the

passive sinking velocity of POC in the water column in m d$^{-1}$, $k$ is the first-order decay rate of POC in d$^{-1}$.

The POC transport model was numerically solved with POC concentration in the center of each box and exchange fluxes

defined on the grid cell interfaces. The flow velocity output from the hydrodynamic model (in 3 hour timesteps) was linearly

interpolated in time to obtain flow velocities at every model integration step. To decrease computation time and close the

mass balances, vertical flow was calculated from the flux divergence of the horizontal flow and assuming a constant free-

surface or zero elevation (η). Comparing the vertical flow from this method with ROMS output vertical flow, the spatial

pattern of vertical velocities compares well (Supplementary Figure A2). The model was numerically integrated using a

variable-order Adams-Moulton predictor-corrector scheme, as implemented in the R-package deSolve (R Core Team, 2019;

Soetaert et al., 2010). Advection was implemented using simple first-order upwind differencing; due to the numerical

dispersion that this method generates, no horizontal or vertical diffusion was imposed and numerical dispersion was

generally low.

### 2.3.3    Benthic cold-water coral biomass and sediment models

We imposed CWC biomass and sediment organic matter at the bottom boundary layer of the model domain (see conceptual

diagram in Figure 2). The CWC biomass model is based on CWC organic matter uptake and physiology. We express CWC

biomass based on the metabolically active tissue, with tissue organic carbon as proxy, and exclude inorganic carbon of their

calcium carbonate skeletons. In our model, CWC take up POC from the bottom layer of the organic matter transport model

by suspension feeding. To take the physical constraints of a coral into account, we use several efficiency parameters and a

CWC surface-to-biomass conversion factor. The net change in CWC biomass ais calculated as logistic growth with a

carrying capacity, a per capita CWC growth rate, and a metabolic cost rate. First, we describe the model equations, followed

by a rationale on the chosen parameters and values. An overview of all parameters and values is also given in Table 1. The

CWC model is described as:



$$\frac{dCWC_b}{dt} = CWC_b \cdot \left(1 - \frac{CWC_b}{CC_{CWC}}\right) \cdot POC_{bbl} \cdot v_{bbl} \cdot A_{CWC} \cdot e \cdot FP_{CWC}(v_{bbl}, k_v, FP_{max}) - m_{CWC} \cdot CWC_b \quad \text{(Equation 2)}$$

Where $\frac{dCWC_b}{dt}$ is the net change in CWC biomass ($CWC_b$) over time (in mmol C m$^{-2}$ d$^{-1}$), $CC_{CWC}$ is the carrying capacity

(6000 mmol C m$^{-2}$), $POC_{bbl}$ is the POC concentration in the bottom boundary layer (mmol m$^{-3}$), $v_{bbl}$ is the bottom boundary

layer current speed (in m d$^{-1}$) calculated as $v_{bbl} = u^2 + v^2$, $A_{CWC}$ is the polyp-surface-to-biomass ratio of *D. pertusum* (7.48 $\cdot$

$10^{-5}$ m$^2$ (mmol C)$^{-1}$), $e$ is the dimensionless feeding efficiency in which a CWC incorporates passing food particles into

biomass and is approximated as the POC capture efficiency (CE$_{CWC}$; 0.15) times the assimilation efficiency (AE$_{CWC}$; 0.80)

and by the net-growth efficiency (NGE$_{CWC}$; 0.10), $FP_{CWC}$ is the fraction of extended polyps (see Equation 3), and $m_{CWC}$ is the

basal CWC respiration rate (0.0035 d$^{-1}$). $FP_{CWC}$ is calculated as:

$$FP_{CWC} = FP_{max} * \left(1 - \frac{v_{bbl}{}^p}{v_{bbl}{}^p + k_v{}^p}\right) \text{(Equation 3)}$$

where $FP_{CWC}$ is the fraction of open CWC polyps fitted on observational data (Figure A3), $FP_{max}$ is the maximum fraction of

extended polyps, as measured in situ by Osterloff et al. (2019), $p$ is a dimension fitting parameter, $k_v$ is the current speed at

which half of the CWC polyps are open in m s$^{-1}$.


The parameters and values are chosen as follows: the surface-to-biomass ratio ($A_{CWC}$) is defined as the biomass-specific

feeding area, or the polyp area, per mmol C of coral and is expressed in m$^2$ (mmol organic C)$^{-1}$. A colony of *D. pertusum*

contains 2.40 ± 0.99 polyps per g dry weight (mean ± SD; n = 15; Gori et al., 2014), and consists for 1.36% ± 0.35 of organic

carbon per g dry weight (de Froe et al., 2019; Larsson et al., 2013; Maier et al., 2019). Therefore, *D. pertusum* comprises of

2.12 ± 0.87 polyps per mmol C. The polyps of *D. pertusum* have a surface area of 35.3 ± 2.0 mm$^2$ polyp$^{-1}$ (Purser et al.,

2010) and multiplying the polyp surface area with the number of polyps per coral biomass gives 7.48 $\cdot$ 10$^{-5}$ m$^2$ per mmol C.

This surface-to-biomass ratio represents the maximum feeding surface, but not all polyps of a colony are always fully

extended, e.g. current speed influences the polyp behavior of corals (Osterloff et al., 2019). At high flow speeds, corals can

retract their polyps to reduce the force drag on their tissue. We implemented this behavior by fitting experimental data of

polyp extension versus current velocity (Equation 3; Figure A3; Orejas et al., 2016b). Additionally, not all extended polyps

are successful in capturing a food particle. A study of a hydroid coral showed that between 5 and 60% of polyps were

successful in capturing a food particle, depending on the flow regime and coral morphology. Here, we introduce this

parameter as $CE_{CWC}$, and as the mean capturing efficiency for an elongated coral colony ranged from 0.10 to 0.30, we set its

value to an intermediate value of 0.15 (Hunter, 1989).




CWCs can also experience intra-colonial polyp competition, i.e. inner polyps of a colony may filter water which was already partially filtered by polyps on the outside of the colony (Galli et al., 2016; Kim and Lasker, 1998). This effect is considered by introducing a carrying capacity ($CC_{CWC}$) or maximum population density. The carrying capacity is described in the benthic model as: $\left(1 - \frac{CWC_b}{CC_{CWC}}\right)$, so coral growth rate approaches zero when the population density reaches the carrying

capacity. The maximum CWC density measured in a boxcore in the study area was 1,800 mmol C m$^{-2}$ in which living corals covered ~20% of the box core (De Clippele et al., 2021a; de Froe et al., 2019). A video transect found maximum coverage of 60% of the seafloor on a large coral mound in the study area (Maier et al., 2021). Based on these data, we set the carrying capacity of CWCs to 6,000 mmol C m$^{-2}$. Food particles captured by corals are divided into a digestible and a non-digestible fraction with the assimilation efficiency ($AE_{CWC}$). Direct measurements of the assimilation efficiency for CWCs are not

available as experimental measurements are often difficult because the non-assimilated part of the food is often mixed with particulate mucus (Wild et al., 2008). However, the release of particulate organic matter (i.e. non-assimilated matter and mucus) is found to be small compared to the food uptake (Maier et al., 2019) and the $AE_{CWC}$ is therefore set to 0.8, consistent with values reported for tropical corals (Anthony, 1999). The assimilated food is used for growth and for the maintenance metabolism. The fraction of carbon incorporated into the tissue compared to the assimilated carbon is known as the net

growth efficiency ($NGE$). Although data on NGE for CWCs is scarce, two studies estimate the $NGE$ for *D. pertusum* to be between 0.05 and 0.3 (Maier et al., 2019; van Oevelen et al., 2016). Therefore, the $NGE_{CWC}$ is set to 0.1, which is low compared to better-studied taxa like zooplankton (>0.5) and shallow-water anemones (0.3 – 0.6; Anderson et al., 2005; Zamer and Shick, 1987). However, a low growth efficiency and slow growth are typical for long-lived species such as *D. pertusum* (Roberts et al., 2009).


The sediment organic matter model is described as:

$$\frac{dPOC_{sed}}{dt} = w_s * POC_{bbl} + (1 - AE_{cwc}) * CWC_{in} - k_{sed} * POC_{sed} \quad \text{(Equation 4)}$$

Where $\frac{dPOC_{sed}}{dt}$ is the change in sediment organic matter concentration over time (in mmol C d$^{-1}$), dPOC$_{sed}$ is the concentration of organic matter in the sediment top layer (in mmol C per m$^{-2}$), $w_s$ is the passive sinking rate of POC in the bottom layer (10 m d$^{-1}$), $POC_{bbl}$ is the organic matter concentration in the bottom layer grid cell in mmol C m$^{-3}$, $AE_{CWC}$ is a dimensionless assimilation efficiency parameter (0.80), $CWC_{in}$ is the organic matter uptake by CWCs in mmol C m$^{-2}$ d$^{-1}$ (i.e. first term in eqs. 2) and $k_{sed}$ is the sedimentary organic matter respiration in d$^{-1}$.






**Table 1: model parameter values, conversion factors, and respective sources**

| Parameter | Description | Value | Unit | Source |
|-----------|-------------|-------|------|--------|
| $F0$ | Water column POC upper boundary export production | 12 | mmol C m$^{-2}$ d$^{-1}$ | (Henson et al., 2015) |
| $K_{poc}$ | Water column POC decay rate | 0.030 | d$^{-1}$ | (Henson et al., 2015; Yool et al., 2011) |
| $w_s$ | Water column POC passive sinking speed | 10 | m d$^{-1}$ | (Riley et al., 2012) |
| $A_{cwc}$ | CWC surface-to-biomass ratio | $7.48 \cdot 10^{-5}$ | m$^2$ mmol C$^{-1}$ | (de Froe et al., 2019; Gori et al., 2014; Larsson et al., 2013; Maier et al., 2019; Purser et al., 2010) |
| $CE_{cwc}$ | CWC capture efficiency, fraction of polyps that capture a food particle | 0.15 | - | (Hunter, 1989) |
| $AE_{cwc}$ | CWC assimilation efficiency, fraction digestible POC | 0.80 | - | (Anthony, 1999) |
| $NGE_{cwc}$ | CWC net-growth efficiency, fraction digested POC used for growth | 0.10 | - | (Maier et al., 2019; van Oevelen et al., 2016) |
| $FP_{max}$ | CWC maximum fraction of extended polyps | 0.90 | - | (Orejas et al., 2016a; Osterloff et al., 2019) |
| $p$ | CWC scaling parameter for calculating fraction of extended polyps related to bottom current speed | 1.5 | - | (Orejas et al., 2016a); this study |
| $k_v$ | CWC flow speed at which half of polyps are rectracted | 0.40 | m s$^{-1}$ | (Orejas et al., 2016a) |
| $CC_{cwc}$ | CWC carrying capacity, maximum biomass density | 6000 | mmol C m$^{-2}$ | (de Froe et al., 2019; Maier et al., 2021) |
| $m_{cwc}$ | CWC basal respiration rate | 0.0035 | d$^{-1}$ | (Larsson et al., 2013) |
| $k_{sed}$ | Sediment respiration rate | $4.1 \cdot 10^{-2}$ | d$^{-1}$ | (Westrich and Berner, 1984) |



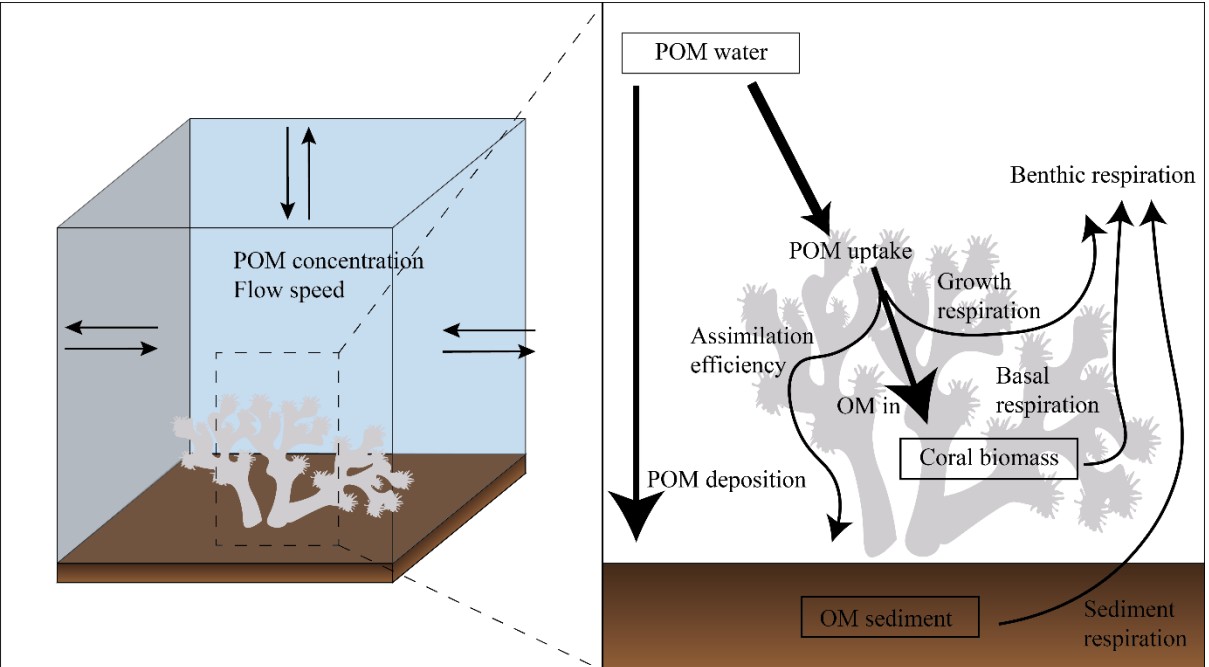

**Figure 2: conceptual diagram of the here developed cold-water coral and sediment model. The left panel shows one bottom grid cell, the arrows indicate water and POC exchange with surrounding grid cells. The right panel represents the benthic cold-water coral and sediment model.**

## 2.4 Procedure to solve the coupled model

Model spin-up of the fully coupled model proved to be computer intensive as CWC growth operates on a time scale of years whereas hydrodynamics operate on timescales of ~ hours. The following three-step procedure for model spin-up was therefore performed (see Figure 3 for a workflow diagram): (1) organic matter transport initialization, (2) CWC biomass and sediment organic matter initialization, and (3) coupled benthic-biogeochemical model runs.

In step 1, the initial suspended particulate organic matter concentration was imposed by an exponential decrease with water depth, reflecting passive sinking of POC (Figure 3A; Martin et al., 1987). Advective transport of organic matter is spin-up by running the organic matter transport model consecutively for three months. After which water column POC concentration remained in more or less steady state in each consecutive run.

The output of step 1 was used to initialize the CWC biomass and sediment POC concentration in step 2 (Figure 3 C&D). The bottom boundary POC concentration ($POC_{bbl}$) and the bottom boundary current speed ($v_{bbl}$) averaged per bottom grid cell over the whole one-month model run were used to calculate initial CWC biomass and sediment POC concentration for each



bottom grid cell (see Equation 2 and Equation 4). This resulted in high CWC biomass along the 500 – 1000 m depth range in the model domain (see Figure 3D inset).

The outputs of step 1 (initial water column POC concentration) and step 2 (initial CWC biomass and sediment POC concentration) are then used as initial conditions for the fully coupled model runs (Figure 3 E). In the coupled model runs, the uptake of POC by CWCs would decrease POC concentration in the bottom boundary layer. The first time we ran the fully coupled model it appeared that step 2 overestimated the CWC biomass, as in the fully coupled model corals depleted POC to such an extent that it led to a rapid decrease in coral biomass throughout the domain (Figure 3 F). CWCs are slow

growing organisms and the chosen physiological parameters in our model therefore only allow CWCs to increase/decrease relatively slowly in biomass in our model domain. For example, if the CWCs were starved and no food would be available, the basal metabolic cost rate (0.0035 d$^{-1}$) would only allow CWC biomass to decrease by 0.35 % per day. As each model run represents one month, it would take considerable amount of time and model runs to reach a (dynamic) biomass equilibrium in the coupled model. To speed up computation, we followed two steps. First, we divided initial CWC biomass from step 2

by three and used that as initial benthic biomass. This new initial CWC biomass would still be higher than CWC equilibrium biomass throughout the model domain, as CWC biomass was still declining in consecutive model runs. Second, we ran the coupled model for a total of five consecutive months with a coral growth/decline enhancement factor of 12. This is a method that is also used in morphological and sediment transport modelling approaches (Ranasinghe et al., 2011). Finally, two months were run without the growth- enhancement to arrive at the final output, in which dCWC$_b$ / dt was close to steady-

state.

## 2.5   Data sources for model-data comparison

To compare our model predictions and evaluate model performance, we compare our model output with recent observational studies in the study area (Figure 4 A & B). Two moorings measured current profiles, turbidity, and fluorescence for a full

year (Schulz et al., 2020). Benthic respiration rates were quantified on two mounds in the Logachev mound province, and an adjacent sediment area (de Froe et al., 2019). Our CWC biomass predictions are qualitatively assessed with observations. The % cover of living CWC, dead framework, and sediment was quantified from video transects and used as a proxy for CWC biomass (De Clippele et al., 2019; Maier et al., 2021). Finally, a statistical habitat suitability model has been developed for the same model domain (Rengstorf et al., 2014), as well as a machine learning regression-based biomass maps

(De Clippele et al., 2021a). The results of these studies are used for model validation and to discuss our findings.





**Figure 3: Workflow diagram describing the steps used to spin up the coupled model with the large panels showing the average POC concentration in the bottom layer across the model domain and the small inset panel showing the cold-water coral biomass**





**across the model domain. 'NA' means that predictions of the CWC biomass are not yet available at that stage. A) initial bottom layer POC concentration based on the Martin's curve Martin et al. (1987), B) bottom-layer POC concentration after spin-up of POC transport by advection, which is used as input (C) for initializing benthic biomass. Initialized cold-water coral biomass used as input (E) to couple the POC transport model with the benthic model. F) POC depletion in the bottom water column after running the coupled model with one week of data. Running the coupled model for seven times results in our final CWC biomass predictions.**






**Figure 4: Observational data locations used to validate the model results. A) Map of whole model domain with dotted lines illustrating the bathymetry and the black line indicates the cross-section over Haas mound used for Video A2. B) expanded detail on where most data was collected. Box cores data is used to validate benthic respiration and biomass, mooring data is used to validate hydrodynamics and POC transport, coral cover data from video transect (VT) are used to validate CWC biomass predictions, "Coral presence" lines are the contours of area within which suitable CWC habitat is predicted by Rengstorf et al. (2014). [1]de Froe et al. (2019), [2] Schulz et al. (2020), [3] Maier et al. (2021), [4] De Clippele et al. (2019), [5] Rengstorf et al. (2014).**






## 3    Results & discussion

Our CWC biomass predictions tend to follow the bathymetry with high CWC biomass predicted on the flanks and summits
of the CWC mounds of which highest values are found on the flanks (Figure 6). Below, we first compare our predictions
with available data on 1) hydrodynamics, 2) POC concentration and transport, 3) CWC distribution, and 4) respiration.
Subsequently, we discuss the distribution of predicted CWC biomass in the model domain, compare our model predictions
with machine learning modelling methods, and discuss the limitations and prospects of our model results.

### 3.1    Comparing model performance with observations

**3.1.1    Hydrodynamic- and organic matter transport model**

Hydrodynamic forcing of our 3D model shows that the general or residual current in the model is directed south-westerly
along the slope of Rockall Bank (Figure A4). This is a result of anti-cyclonic circulation circumventing Rockall Bank (e.g.,
Ellett et al., 1986; Holliday et al., 2000; Johnson et al., 2010). The area is subject to a dominant diurnal tidal system where
barotropic diurnal tidal waves are trapped, causing cross-slope transport with a diurnal periodicity (Huthnance, 1973;
Pingree and Griffiths, 1984). These tidal currents cause high vertical velocities on the flanks and summits of the coral
mounds following a diurnal and spring-neap tidal cycle (Figure A2 B; Mohn et al., 2014). The hydrodynamic forcing is
validated in Mohn et al. (2023) with current velocity data from various moorings in the region (see Schulz et al. (2020) and
White et al. (2007)). The general direction of POC transport in our model domain was in southwest direction along Rockall
Bank slope (Video A1) as also measured by Schulz et al. (2020), while tidal currents cause POC to be transported north-
south over a diurnal tidal cycle (Video A1). The vertical currents above the CWC mounds caused episodic transport of POC
towards the seafloor (Figure 5; Video A2), a process which was observed in earlier modelling (Soetaert et al., 2016a) and
observational studies (Duineveld et al., 2007; de Froe et al., 2022).

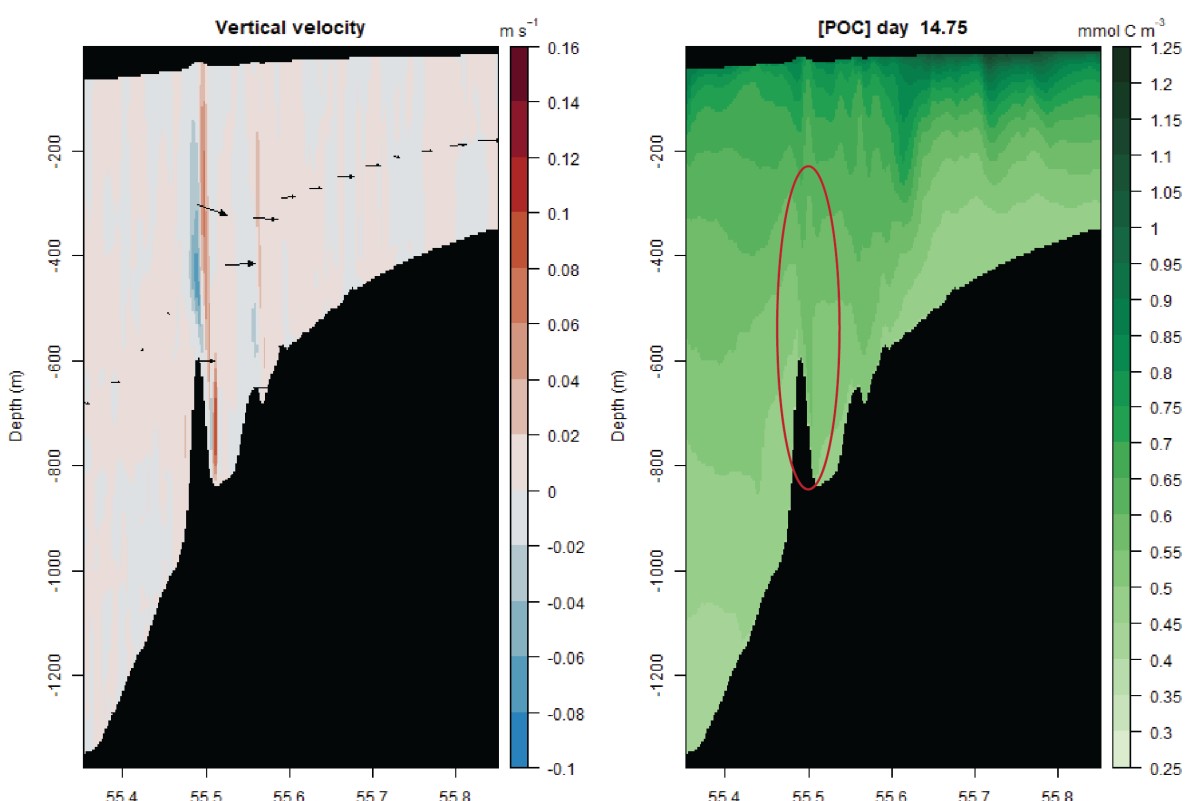

**Figure 5: model output of vertical velocity (left panel) and water column POC concentration (right panel) along the Haas transect (Figure 4). The red ellipse indicates the POC increase close to the mound.**

### 3.1.2    Cold-water coral biomass- and sediment model

The CWC biomass was predicted to be highest on the summits and south- and southwestern upper flanks of the CWC mounds (Figure 6). Overall, our model results match data of CWC/dead framework cover along video transects (De Clippele et al., 2019; Maier et al., 2021), as highest CWC biomass was predicted in areas where high cover of live CWCs/dead coral framework was observed and low CWC biomass was predicted in areas with high sediment cover (Figure 7; Figure 8). However, how well model-observational data matched differed between video transects. For example, transect VT7 showed

a large discrepancy between modelled and observed CWC biomass/cover (Figure 8 F) i.e., high coral cover predicted in areas where high sediment cover was observed. This mismatch could have several causes: first, the horizontal resolution (250 m) of the model caused differences in model depth and video transect depth. For example, the depth of the model domain significantly differed compared to the observed depth in video transects 6 and 7 (Supplementary Figure A5), caused



by the different spatial resolutions between model (250 m) and video transects (1 m). Second, due a patchy distribution of
CWCs and the limited range of view (1-2 m wide) of video cameras, the video transects may have missed the presence of
live CWC. Nevertheless, our modelling results show that the distribution of CWCs can be predicted on a regional scale
based solely on coarse local hydrodynamics and organic matter transport, strengthening the hypothesis that a steady food
supply or influx is a key driver for CWC growth and occurrence (De Clippele et al., 2021b; Hebbeln et al., 2019; Maier et
al., 2023; Portilho-Ramos et al., 2022).


Although only a qualitative comparison between video data and model predictions can be made due to the different units that
are used for biomass quantification, interesting observations surface from this comparison. The modelled CWC biomass
distribution show good agreement with cover of living corals, but predictions show better agreement with the observed cover
of live CWC and dead coral framework combined (Figure 7, Figure 8). Dead coral framework is considered CWC skeleton
without a live coral tissue, that forms habitat for many associated fauna and microorganism species (Freiwald and Wilson,
1998), which together account for a substantial part of benthic respiration (de Froe et al., 2019) and a complex nitrogen cycle
(Maier et al., 2021). CWCs grow in patches (Wilson, 1979), where the living polyps extend their tentacles in the water
column to feed on suspended particles, and the polyps closer to the seafloor eventually die-off and become buried by baffling
of sediment (Mienis et al., 2009b; Roberts et al., 2009). The good agreement of our predictions with observational data on
live CWC and dead framework combined could have several causes: first, the difference between the timescale of our model
and CWC reef dynamics. Our model is based on one-month of hydrodynamic model output, organic matter transport and
CWC physiology. CWC biomass can decrease in the model domain due to the basal CWC respiration rate, but mortality or
longevity of CWCs is not included. This means that, in our model, if conditions remain favorable CWCs can exist
indefinitely. Although, generally little is known on the temporal and spatial dynamics between living CWCs and dead coral
framework on a reef, CWCs would die-off at one point in time and become dead coral framework. Second, it could be that
CWCs have grown in the past in the areas where we predict high CWC biomass, but which have died-off due to conditions
that were not included in our model (i.e., infection, predation, temperature, ocean acidification). Therefore, it is reasonable to
find dead coral framework where high CWC biomass is predicted in our model. The presence of dead coral framework on
the mounds indicate areas that were favorable for CWC growth in the past. It would be interesting to expand our CWC
biomass model with dead coral framework as a state variable, where dead coral framework is build up with a mortality rule
(as in Hennige et al. 2021). This would especially be interesting as dead coral framework keeps affecting hydrodynamics
(Bartzke et al., 2021; Corbera et al., 2022) and sediment baffling (Wang et al., 2021) through its structure, and nutrient
cycling through the community associated with dead coral framework (Maier et al., 2021).

Modelled CWC biomass, ranging from 50 – 850 mmol C m$^{-2}$ on the CWC mounds and close to zero in the sediment areas of
the study area (Figure 6), was on the lower side of observed CWC biomass (e.g., ~ 1850 mmol C m$^{-2}$; de Froe et al.,



2019) and estimates from a machine learning model (~ 4200 - 5237 mmol C m$^{-2}$; De Clippele et al., 2021a). Modelled benthic respiration ranged between 3.7 and 40.9 mmol C m$^{-2}$ d$^{-1}$ and closely followed the CWC biomass spatial distribution (Figure 9A), indicating that CWCs are largely responsible for the benthic respiration (up to 70%) in areas with high CWC biomass (Figure 9B). Sediment-based respiration was enhanced in areas with high CWC biomass (Figure A6), due to organic matter deposition on the underlying sediment and a higher POC concentration in the bottom layer. Modelled benthic respiration in the model domain compared generally well with observational data (de Froe et al., 2019; Figure 9C), and the modelled benthic respiration on coral mounds (10 – 40 mmol mmol C m$^{-2}$ d$^{-1}$) was comparable with observational data from CWC reefs in the northeast Atlantic (7.7 - 122 mmol O$_2$ m$^{-2}$ d$^{-1}$ Cathalot et al., 2015; Khripounoff et al., 2014; Rovelli et al., 2015; White et al., 2012). Assuming that modelled benthic respiration was representative throughout the year, the model domain seafloor would respire in total 104,845 tonnes C per year. CWCs alone were responsible for 11,260 tonnes C yr$^{-1}$ of benthic respiration, or 10.7% of the total benthic respiration in the model domain, while only on 2.8% of the model domain, a CWC biomass >100 mmol C m$^{-2}$ was predicted. Our predicted CWC-based respiration was comparable to, but at the upper end of the carbon turnover estimate of 5,763 to 9,260 tonnes C yr$^{-1}$ which was predicted from a CWC suitable habitat model (De Clippele et al., 2021; Rengstorf et al., 2014).

The good correspondence between modelled and observational benthic respiration, but low biomass estimates in the model compared observations can be explained by the chosen parameter values for our CWC biomass model. There are several parameters that are currently poorly constrained. CWC carrying capacity was, for instance, estimated based on box cores retrieved in de Froe et al. (2019), but a maximum CWC density on a reef is currently not resolved. Some physiological parameters for CWCs, such as 'assimilation efficiency' ($AE_{CWC}$) and 'net growth efficiency' ($NGE_{CWC}$), were presently only constrained based on a few studies. It could be that our carrying capacity (6000 mmol C m$^{-2}$) estimate was an underestimation, and the values chosen for $AE_{CWC}$ (0.80) and $NGE_{CWC}$ (0.10) were an overestimation. If we would set the carrying capacity higher but also decrease values for $AE_{CWC}$ and $NGE_{CWC}$, CWC biomass would be increased in the model while benthic respiration would stay at similar levels.



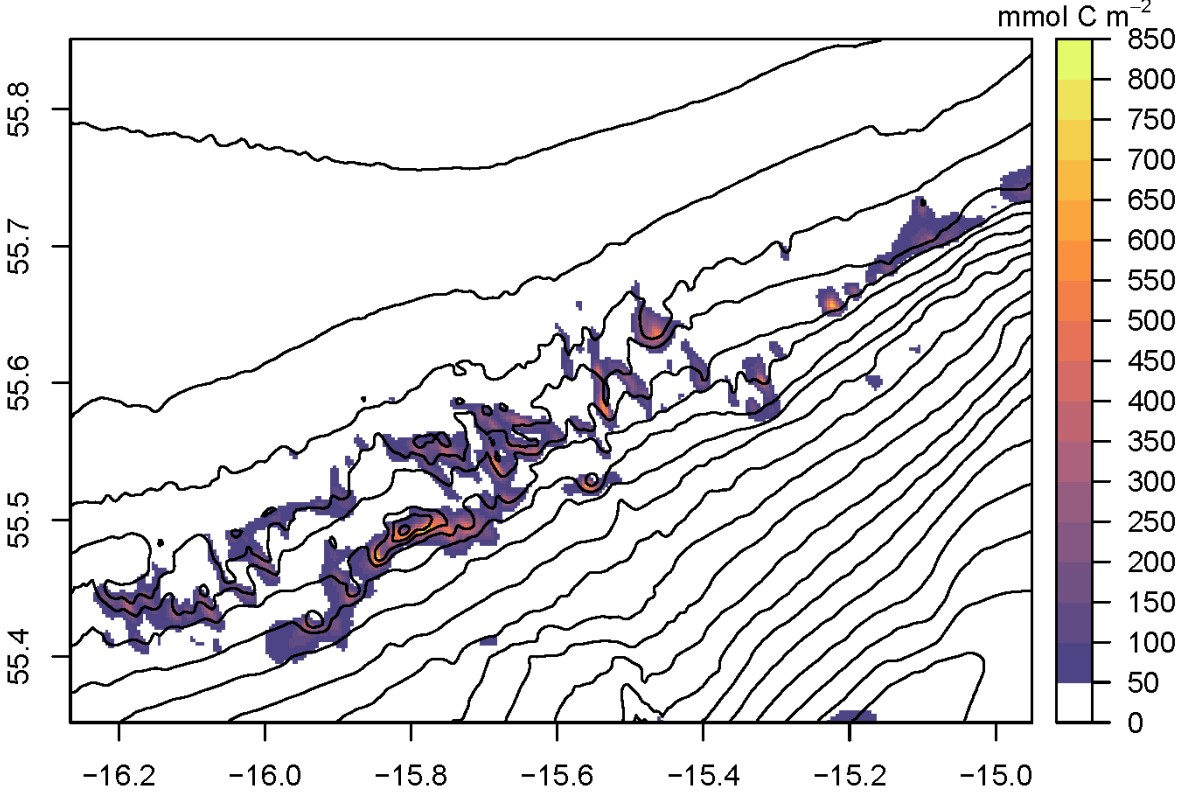

**Figure 6: predicted CWC biomass in mmol C m$^{-2}$. Black lines indicate the bathymetry of the model domain (range in depth = 300 – 1900 m).**





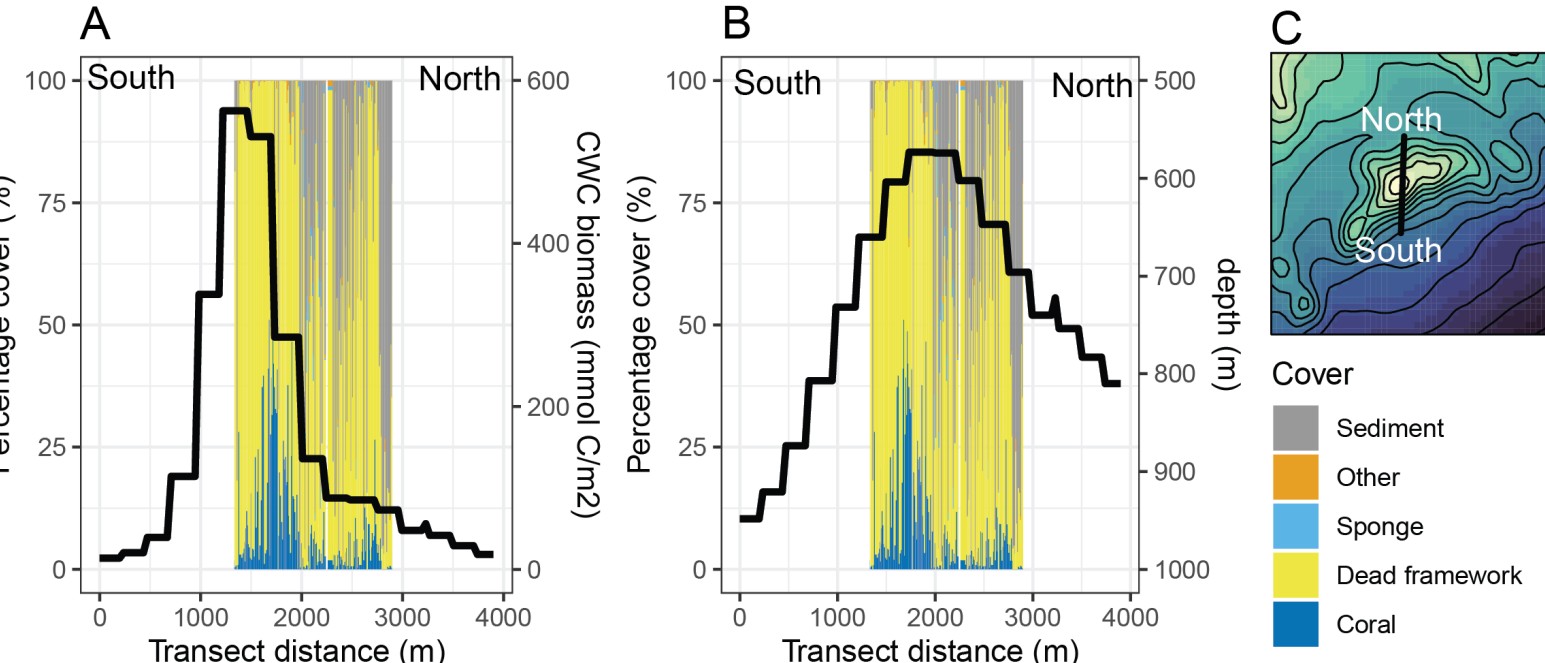

**Figure 7: Comparison between model CWC biomass predictions and video transect data percentage cover data (Video 7, Figure 4) from Maier et al. (2021). Cover is in percentages, category "Other" includes other macrofauna species. A) predicted CWC biomass indicated by the black line and the scale on the right Y-axis (in mmol C / m², B) depth of model domain (in m) along the video transect indicated by the black line and scale on the right Y-axis, C) Location of video transect over Haas mound.**





**Figure 8: Comparison between model prediction and video transect percentage cover data (VT2 – VT7, Figure 3) from De Clippele et al. (2019). Model predictions are indicated by black line and biomass scale is on the right Y-axis. A) VT2, B) VT3, C) VT4, D) VT5, E) VT6, F) VT7, G) locations of the video transects in the model domain (see also Figure 4). The number one on the map (G) indicates starting point of each transect.**






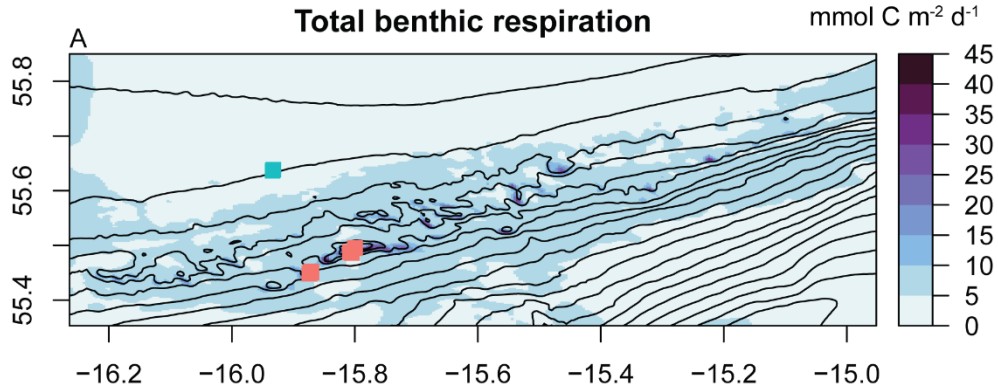

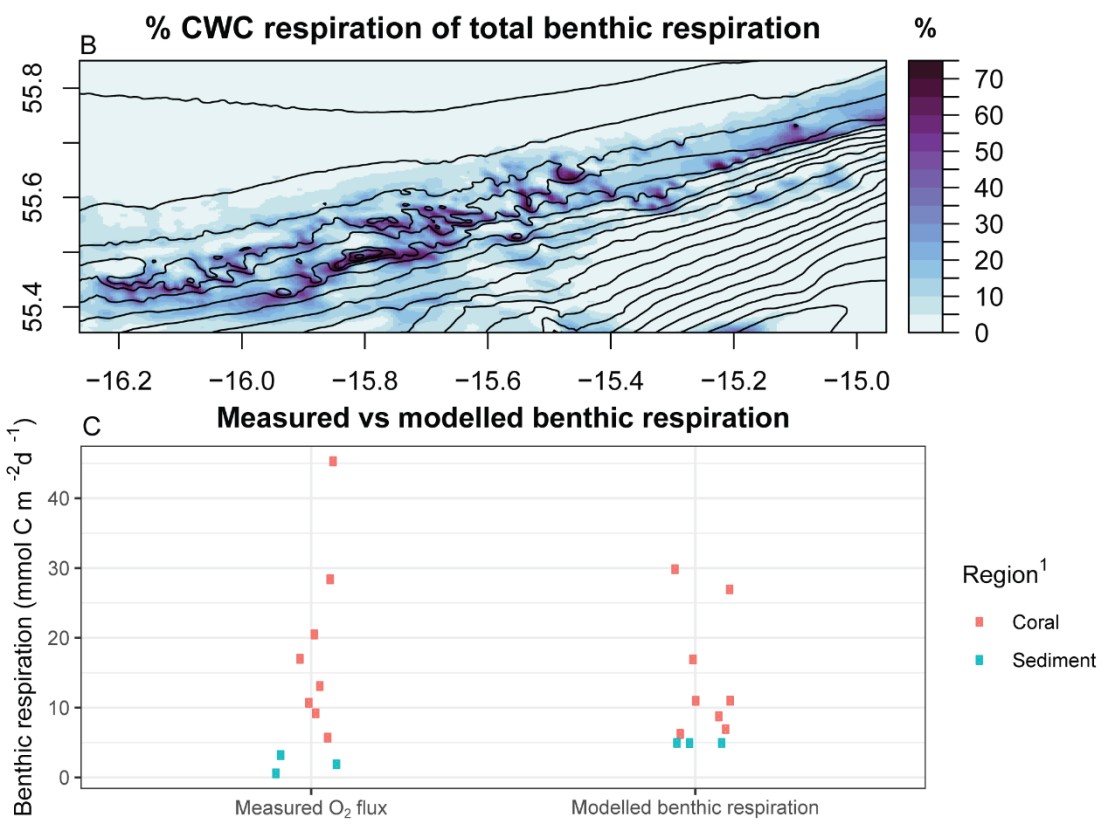

**Figure 9:** Modelled benthic respiration compared to observational data from de Froe et al. (2019). A) total benthic respiration and measurement locations indicated in coloured points, B) percentage of total benthic respiration caused by cold-water coral respiration, C) measured versus modelled benthic respiration. [1] regions are based on de Froe et al. (2019) and Rengstorf et al. (2014).





**3.2    Coupling the benthic- with the pelagic model is key in predicting cold-water coral density**

The here presented three-step modelling approach (Figure 3) provided insight into what drives CWC density in our model domain. Predicting CWC biomass without coupling the benthic with the pelagic model (step 2; section 2.4), resulted in an overestimation of CWC biomass (Figure 3 D inset). Coupling the pelagic POC model with the benthic CWC/sediment model (step 3; section 2.4), resulted initially in strong depletion of POC in bottom waters, except above the coral mounds and

ridges (Figure 3F). In areas where POC was depleted in bottom waters, CWC respiration (right term Equation 2) was higher than food uptake (left term Equation 2) and CWC density decreased in those areas. On the coral mounds and ridges, tidally induced vertical currents transported POC towards the seafloor and replenished bottom water POC (Figure 5; Soetaert et al., 2016b). This lead to CWC growth on the flanks and summits of the mounds and ridges. Although CWCs benefit from strong bottom currents (Hebbeln et al., 2016; Purser et al., 2010; White et al., 2007), our results show that without replenishment of

food particles, bottom waters would quickly be deprived of food particles and CWC biomass would be much lower or would even disappear. These findings suggest vertical transport of POC is vital for CWC growth and support the idea that tidal dynamics are crucial for sufficient food supply to the CWC reefs on these mounds (de Froe et al., 2022; van Haren et al., 2014; Juva et al., 2020; Soetaert et al., 2016a).

**3.3    Spatial distribution of cold-water corals related to hydrodynamics and organic matter transport**

The south-westerly residual along-slope current and tidally induced cross-slope currents cause POC to be advected up- and down the slope between 500 – 1000 m depth in the model domain, while the net POC transport is in a southwest direction (Video A1). The combined effect of these currents is that POC concentration is relatively frequently replenished above the CWC mounds summits and subsequently transported along their south- and southwestern flanks (Figure 10 A). Under these favorable conditions, high CWC biomass is predicted on the southwestern upper flanks of the CWC mounds (Figure 6,

Figure 10). In addition, westward bottom currents that encounter a mound or ridge are directed southward due to Coriolis force (Figure 10A; Supplementary Figure A7), which further promotes CWC growth on the southern flanks. Furthermore, north of Haas mound we found an area with low bottom current speed where suspended/organic matter is trapped in an eddy circulation (Supplementary Figure A7). This enhances sediment deposition and compares well with the sediment infill found in this area (Mienis et al., 2006).


Besides that CWCs were mostly predicted on the south- and southwestern flanks of the CWC mounds, highest biomass was found specifically close to the summits and on the upper side of the flanks. This pattern was caused by depletion of POC in the bottom layer by CWCs which decreased the quantity of available food for CWCs located downstream (Figure 10B). Bottom water POC depletion by filter-feeding CWCs has also been observed in the field (Lavaleye et al., 2009; Wagner et

al., 2011), and can be interpreted as a negative scale-dependent feedback on CWC growth (van der Kaaden et al., 2020), a mechanism derived from the theory of spatial self-organization (van de Koppel et al., 2005; Rietkerk and van de Koppel,





2008). This mechanism of bottom-water POC depletion proved to be key in successfully predicting CWC biomass. Hence, we argue that the spatial distribution of CWCs in our study area is shaped by a combination of current dynamics, food supply, and scale-dependent feedbacks.


Our mechanistic modelling approach and CWC biomass predictions could also be used to examine CWC mound development and morphology. Although CWC growth and mound development operated on different timescales and our model does not include sedimentation or baffling of sediment, our CWC predictions can indicate in which direction a mound will likely develop. The presence of CWC framework promotes mound development by providing structure and baffling of

sediment (Dorschel et al., 2007; Mienis et al., 2007; Titschack et al., 2015). CWC mounds are globally found in a wide variety of shapes and relative orientation to the general current direction. For example, CWC mounds can be shaped parallel (De Clippele et al., 2017; Hebbeln et al., 2014; Matos et al., 2017) or perpendicular (Correa et al., 2012) to the general current direction. Furthermore, the distribution of CWCs can differ greatly per mound and region. CWCs are found facing the current direction (Buhl-Mortensen et al., 2012; Correa et al., 2012) but are also found on summits, flanks and leeward

sides of CWC mounds (Conti et al., 2019; Dorschel et al., 2007; Lim et al., 2017). The CWC mounds in our model domain are mostly elongated in shape, perpendicular to the southwest-directed net current direction, except for Haas mound, which has its elongated shape parallel to the current direction, and was formed on a pre-existing hump (Mienis et al., 2006). Our CWC biomass predictions fit well with the mound morphologies in the model domain. For instance, in the northwest of the model domain, the CWC ridges shaped perpendicular to the general current direction show high CWC biomass on the

southern side (Figure 10 C). This agrees well with findings from White et al. (2007), who found that CWC mounds are often aligned with the major axis of the tidal current oscillation. Our findings indicate that these ridges will develop further in this direction and might provide an explanation to why these CWC mounds are shaped perpendicular to the current and slope.

### 3.4    Mechanistic cold-water coral predictions compared to statistical methods

The spatial distribution of the here presented CWC biomass predictions align relatively well, but are somewhat shifted

southwestward with the area of > 0.9 probability of CWC presence predicted with a habitat suitability model (Figure 10 D; Rengstorf et al., 2014). Our mechanistic modelling approach allows to dynamically predict CWC growth and provides insight into the mechanisms that drive the spatial distribution of CWCs in the deep-sea. In contrast, habitat suitability models use static data on environmental conditions and terrain variables to predict spatial distribution of CWC cover or presence/absence. Hydrodynamic (non-static) variables such as current speed can be included into habitat suitability models,

which improves model performance (Bargain et al., 2018; De Clippele et al., 2017; Lim et al., 2020; Pearman et al., 2020; Rengstorf et al., 2014). However, we show that bottom current speed alone has limited power to predict habitat suitability as the spatial distribution of CWCs also depends on bottom-water organic matter (POC) concentration and replenishment, which, in turn is affected by CWC growth itself. Recent work in mapping CWC biomass by a combination of habitat suitability modelling and field measurements (De Clippele et al., 2017, 2021a), identified bathymetric position index (BPI)



as most important predictor for CWC biomass in our study area and predicted that CWC biomass was highest on the
summits and crests of the mounds and ridges (De Clippele et al., 2021a). As this compares well with our predictions, BPI,
the relative height of an area compared to its surroundings, might be a good proxy for areas where CWCs experience little
competition for resources. However, as CWC reefs are elevations on a mound slope, the presence of a CWC would also
enhance BPI. Our findings emphasize the necessity to consider feedbacks between CWC growth and resource availability
when predicting spatial distributions of CWCs in the deep sea.









**Figure 10: A)** General transport direction of organic matter in the bottom layer above Haas mound, with colour overlay representing averaged vertical velocity in bottom layer, black lines indicate the bathymetry, and red lines show the predicted flow paths of neutrally buoyant particles that were released on the summits. **B)** POC depletion by CWCs, colour overlay represents averaged POC concentration in the bottom layer, black lines show predicted CWC biomass, and red arrows show the general current direction in the bottom layer. **C)** Detail of CWC predictions on the ridges in the northeast of the model domain. **D)** comparison between CWC predictions and the coral region of Rengstorf et al. (2014) indicated by the red line, **E)** location of subpanels A and C in this figure.

### 3.5    Implications, limitations, and future work

This study presents the first mechanistic modelling approach to predict CWC biomass based on organic matter transport and hydrodynamics. As deep-sea research is restricted by, among other, ship-time, access to infrastructure and equipment (e.g. working class ROVs) the model predictions provide a good means to investigate the spatial distribution and driving mechanisms of deep-sea ecosystems, such as CWC reefs. We could demonstrate that a mechanistic 3D coupled transport-reaction-model successfully predicts CWC biomass. CWC distribution on the south-eastern slope of Rockall Bank is likely driven by the horizontal bottom water POC flux, bottom boundary layer POC replenishment, and spatial competition between CWCs for food. In particular the depletion of organic matter in the bottom water layer by CWCs themselves was key to adequately predict their biomass. The good fit between video observations, benthic respiration data, and modelled CWC biomass strengthens the hypothesis that food supply is the prime predictor for CWC growth (Hebbeln et al., 2019; Maier et al., 2023) within their environmental niche (e.g. temperature and salinity ranges; Dullo et al., 2008; Rüggeberg et al., 2011).

The CWC biomass predictions compare reasonably well with video observations but there are several limitations to the model that should be considered. First, we only used one month of hydrodynamic data to spin-up and run the model, while seasonal variability in environmental/hydrodynamic conditions would affect CWC growth (van der Kaaden et al., 2021; Maier et al., 2020). Second, the metabolic cost of reproduction for CWCs (Brooke and Järnegren, 2013; Maier et al., 2020) is not included in the CWC model, which likely results in CWC biomass overestimation. Third, CWC framework has complex interactions with bottom hydrodynamics which shape the architecture of the CWC framework (Bartzke et al., 2021; Corbera et al., 2022; Sanna et al., 2023) and change the food supply towards the reefs (Guihen et al., 2013; Mienis et al., 2019). Bottom current – CWC framework interaction was not included in the model as it would require resolving spatial scales much smaller than 250 m (horizontal resolution of one grid cell in this model = 250 $m^2$). Finally, resuspension of sedimentary POC concentration is not included in our model, while it increases POC concentration in the benthic boundary layer (Adams and Weatherly, 1981), and increases food supply to CWCs (Mienis et al., 2009a). Incorporating these feedbacks might benefit CWC biomass predictions, but would come at a considerable computational cost, as hydrodynamics would need to be resolved at a much higher spatial resolution.



This modelling approach provides a framework to study the effect of changing temperatures, currents, pH, and nutrients on the spatial distribution of CWCs in the deep sea. For example, the potentially severe effect of rising temperatures and ocean acidification on CWC growth, survival, and accordingly distribution (Chapron et al., 2021; Gómez et al., 2018, 2022; Gori et al., 2016; Guinotte et al., 2006; Lunden et al., 2013; Orr et al., 2005), could be studied by, for example, coupling CWC
respiration to water temperature or mathematically simulating the higher cost of calcification under ocean acidification by increasing basal CWC respiration (Dodds et al., 2007; Gori et al., 2016; McCulloch et al., 2012). In addition, the export of surface-produced POC to the deep-sea could be reduced with climate change (Bopp et al., 2001), which may decrease biomass of CWC reefs and deep-sea benthos in general (Jones et al., 2014; Puerta et al., 2020; Smith et al., 2008). Our model could be used to examine this effect by reducing POC export from the upper boundary layer/condition. Finally, with the
prerequisite of having sufficient physiological data available, our model could also be applied to other suspension feeders. This modelling approach could also be applied in other deep-sea areas, given hydrodynamic data is available (e.g., ROMS-AGRIF output). In summary, our study improves the understanding of the mechanisms driving the spatial distribution of CWCs in the deep sea and provides a tool to investigate this under changing oceanic conditions.

## 4   Conclusion

We investigated drivers of the distribution of CWC biomass by developing a first mechanistic model, based on hydrodynamics, organic matter transport, and CWC physiology. Our model approach successfully predicted CWC biomass on the coral mounds and ridges of the south-eastern slope of Rockall Bank, northeast Atlantic Ocean. High CWC biomass was predicted in regions with strong bottom currents and sufficient replenishment of bottom water organic matter. Benthic respiration on the CWC mounds is mostly driven by CWCs. Coupling the pelagic organic matter transport model with the
benthic CWC biomass and sediment model proved to be key for predicting CWC biomass. This model can be used as a tool in future work investigating the effect of changing ocean conditions on the spatial distribution of CWCs or other suspension feeders in the deep sea.





**5    Appendix A**

**5.1    Figures**

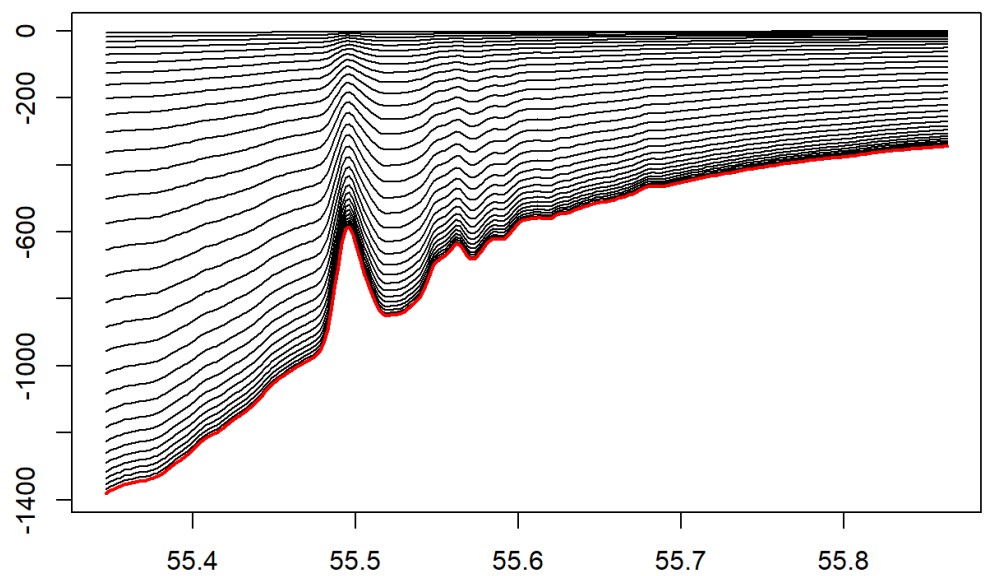

**Figure A1: Latitudinal cross-section of model sigma grid over Haas mound. Example of sigma grid.**



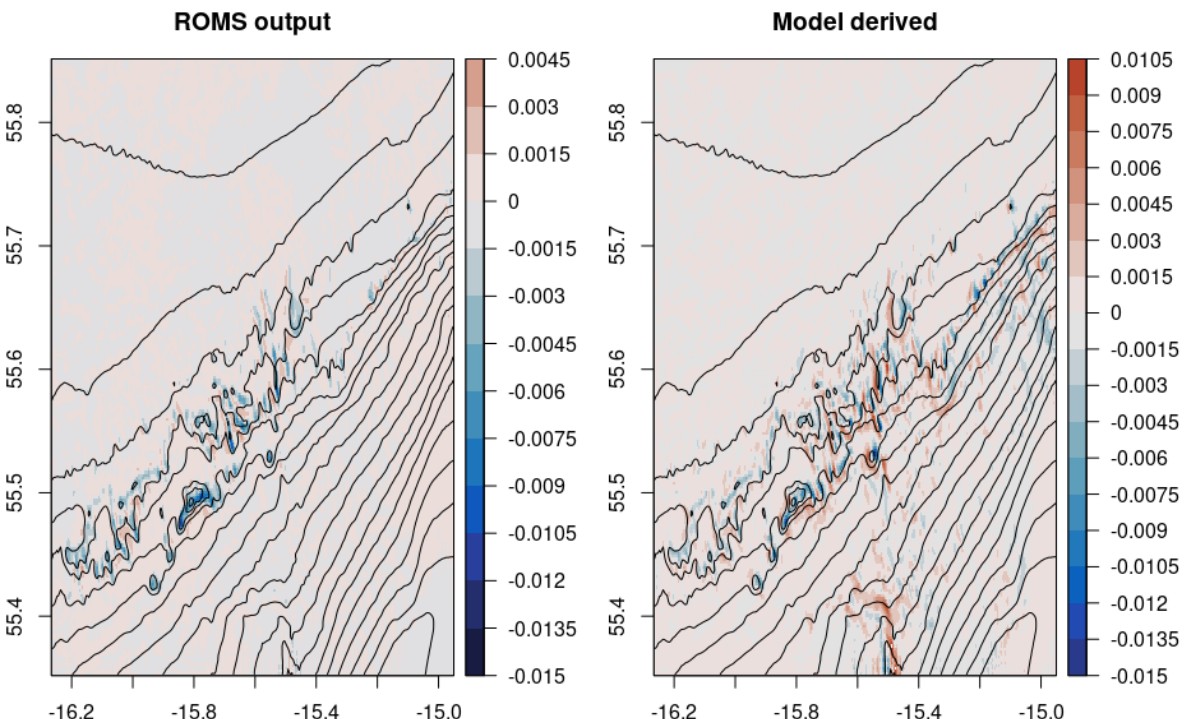

**Figure A2: water column averaged vertical velocities (A) from ROMS output and (B) calculated from the horizontal currents.**




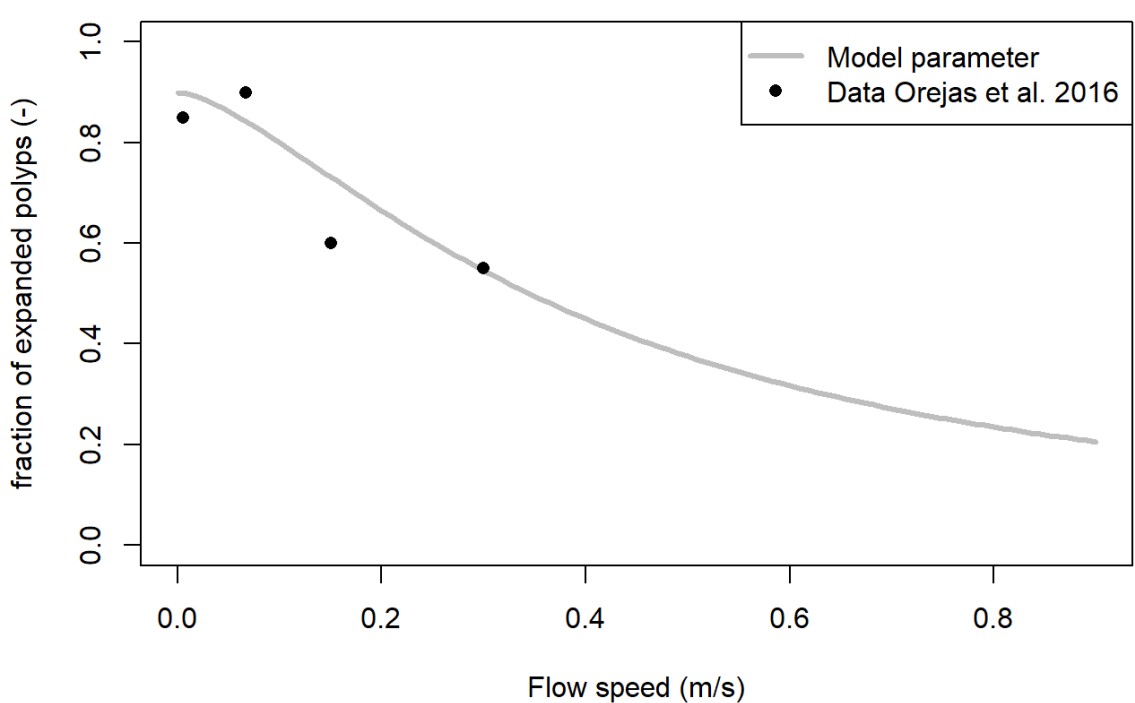

**Figure A3: Fitting of the model parameter FP$_{CWC}$ on data from (Orejas et al., 2016b). The higher the flow rate, the lower fraction of polyps are extended in the Desmophylum pertusum colony.**





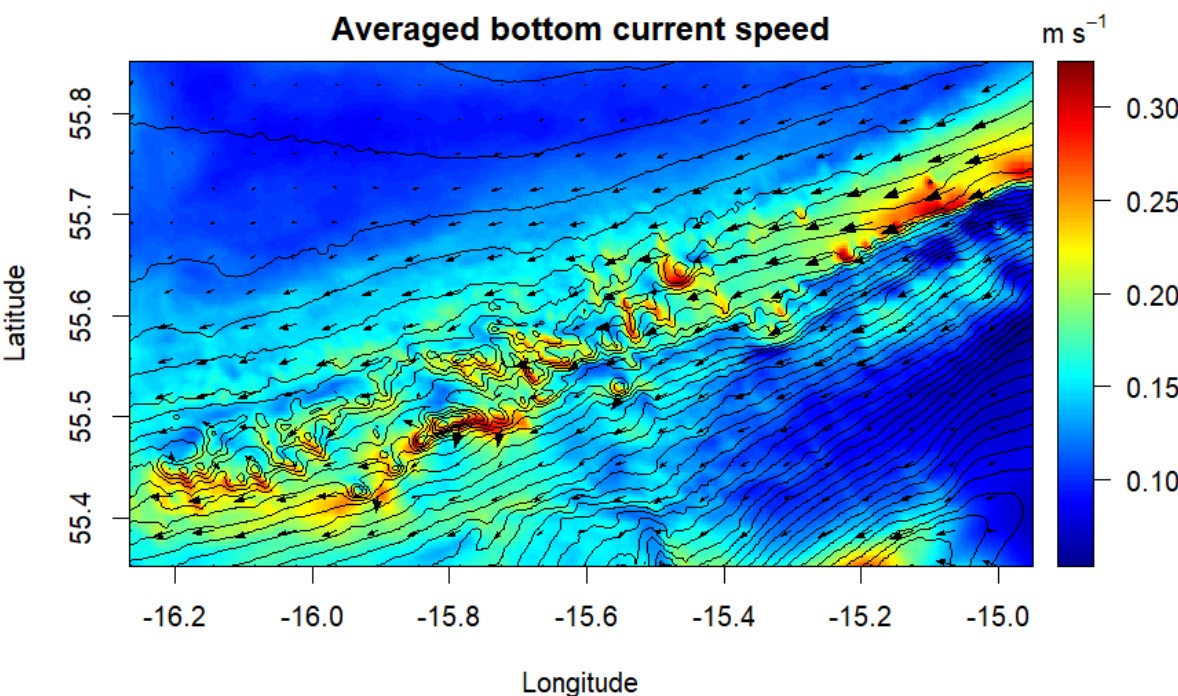


**Figure A4: Averaged current speed of bottom layer and direction.**





**Figure A5: comparison between model and video observations from De Clippele et al. (2019) with A) model depth, B) video transect depth, and C) a map of the transect locations.**



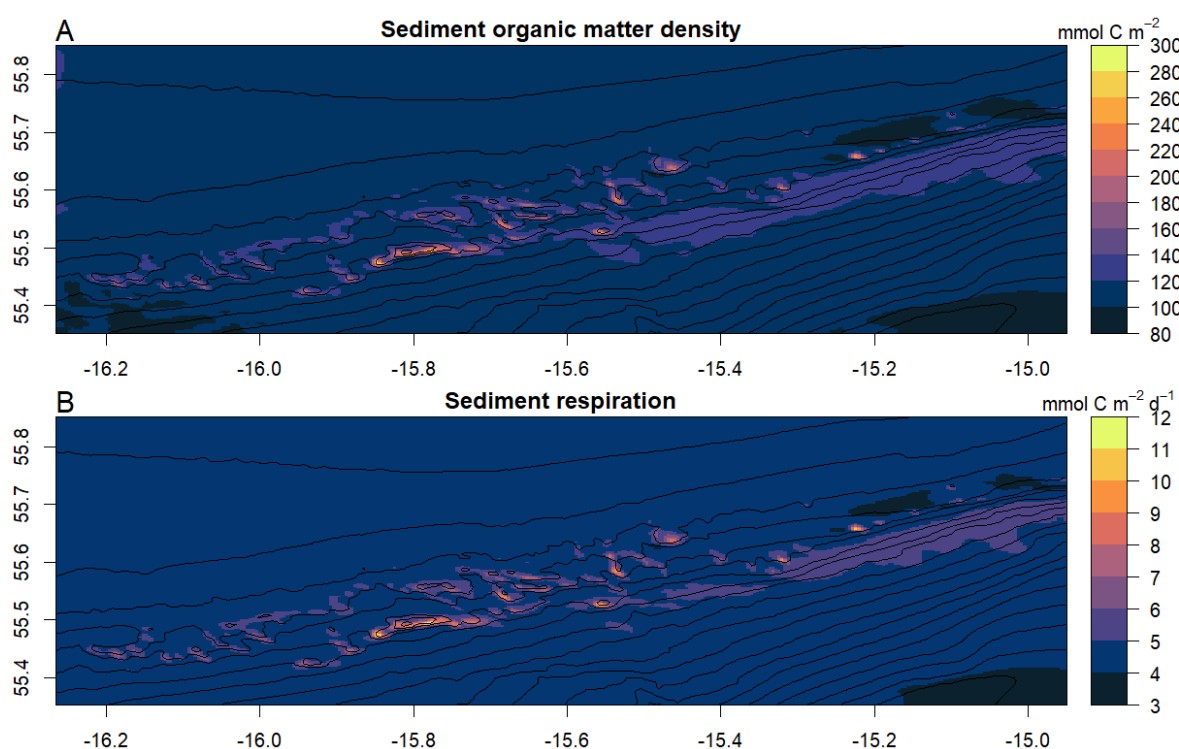

**Figure A6: A) sediment organic carbon density or biomass, B) sediment respiration.**

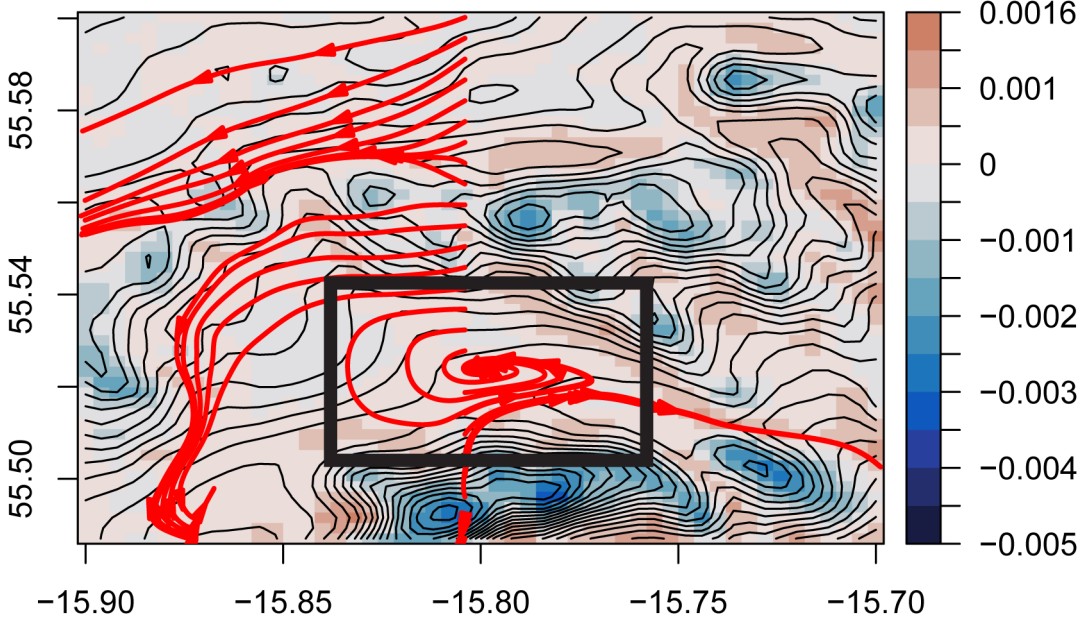

**Figure A7: general direction of organic matter in the bottom layer released along a latitudinal cross-section. Black box indicates where organic matter follows cyclonic circles. The colours represent upward (red) or downward vertical velocity in m s$^{-1}$.**



## 5.2 Videos

**Video A1: depth averaged particulate organic carbon concentration in the water column. Note the overall southwestward direction of the particulate matter, and the north-south sloshing by tidal currents.**


**Video A2: vertical velocity and particulate organic carbon concentration over the Haas mound cross-section. The arrows indicate the north-south direction of the current.**

## 6 Code availability

Model code can be downloaded from: (de Froe et al., 2020)

## 7 Data availability

Model output and input data can be found at: (de Froe et al., 2020). Video transect data were published earlier in (De Clippele et al., 2021a, 2023; Maier et al., 2021).

## 8 Video supplement

Supplementary videos can be found at: (de Froe, 2023)

## 9 Author contribution

EdF, CM, KS, and DvO were involved in the conceptualization of the study. The work was supvervised by DvO, and KS. The model was developed by EdF, CM, KS, AvdK, and DvO. Data was analysed and compared with observations by EdF, GJR, LDC, SM, AvdK, and DvO. All authors commented and wrote parts of the manuscript.

## 10 Competing interests

No competing interests.

## 11 Disclaimer

The output of this study reflects only the authors' view, and the European Union cannot be held responsible for any use that may be made of the information contained therein.

## 12 Acknowledgements

We would like to thank Adri Knuijt from NIOZ for his help with the RStudio server.



## 13    Financial support

This research has been supported by the European Union's Horizon 2020 Research and Innovation programme under grant agreement nos. 678760 (ATLAS) and 818123 (iAtlantic). This output reflects only the authors' view, and the European Union cannot be held responsible for any use that may be made of the information contained therein. Dick van Oevelen and Sandra R. Maier were partly supported by the Innovational Research Incentives Scheme of the Netherlands Organisation for Scientific Research (NWO) under grant agreement no. 864.13.007. Evert de Froe was partly supported by the ArcticNet Network of Centres of Excellence, "Glacier troughs as biodiversity and abundance hotspots in Arctic and subarctic regions" project, ArcticNet Phase V (Geoffroy et al., 2022).

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
