# Peer review of "A first predictive mechanistic model of cold-water coral biomass and respiration based on physiology, hydrodynamics, and organic matter transport"

_EGUsphere, 2025_

## Author Comment (AC1)

**Reply to reviewer 3:**

General comments

de Froe et al., present a very interesting and novel study using mechanistic modelling to predict coral biomass, and respiration. The work is highly interdisciplinary, effectively integrating physical oceanography, ecophysiology, and ecology. To my knowledge, this is one of the first mechanistic models applied to a deep-sea benthic species, and among the first to incorporate feedback loops between environmental and biological processes in deep-sea ecological modelling. The study is therefore both original and of high scientific quality. The materials and methods are clearly described and concise, and the authors demonstrate a strong understanding of coral feeding biology and ecophysiology, which are well integrated into the model. Model limitations are also well explained.

I suggest that the authors emphasize the innovative aspects of their work more strongly. A brief paragraph in the introduction about the use of mechanistic models in deep-sea benthic ecology would help set the context for non deep-sea specialists. Additionally, some key advantages of the approach could be highlighted further. For example, this mechanistic model can estimate parameters that are difficult to obtain with statistical modelling such as species distribution models (e.g., biomass, respiration), especially in data-limited environments such as the deep sea. Statistical models often require extensive datasets and in the case of deep-sea benthic species they are typically limited to presence/absence or abundance data, with very few exceptions, such as the models used for validating the present study. Emphasizing that this model integrates physiological and environmental information without requiring large-scale sampling, except for validation, would strengthen the paper's contribution and practical significance.

Thank you for your kind words and your review. Following your suggestions, we adjusted the introduction by adding a paragraph in the introduction that explains the main difference between mechanistic modelling and statistical modelling approaches, and we also include a few sentences about how mechanistic modelling can be beneficial for deep-sea ecology in general.

In the discussion, section 3.4, we added a few sentences that highlighted the key advantages of mechanistic models for deep-sear research and cold-water coral research. Thereby mentioning that mechanistic models integrate physiological and environmental data. See also the reply on the general comments of reviewer 2. ##

Specific comments

Line 22: A comma after species would help the flow of this sentence.

Done, thank you.

Lines 32-33: Consider moving this sentence up, as it presents the main finding. The authors can then note that it aligns with previous model predictions.

Agree, moved the sentence before the comparison.

Lines 86-92: These lines focus mostly on POC concentration, but the connection to stressors such as acidification or temperature is not straightforward. I suggest adding 1–2 sentences emphasizing the importance of feeding in CWC ecology (e.g., ability to rapidly exploit food pulses whenever they occur, occurrence of seasonal cycles of growth and reproduction that correlate with food availability), and conclude by noting that feeding has been shown to influence responses to climate stressors (e.g., Büscher et al., 2017).

J. V. Büscher, A. U. Form, U. Riebesell, Frontiers in Marine Science. 4, 101 (2017).

Thank you for this remark. In this paragraph we tried to explain that the interaction between a CWC reef and the surrounding environment (food uptake, altering currents) is vital in predicting CWC distribution and we tried to introduce the topic of scale-dependent feedback. Indeed, other environmental conditions also play a role in determining the spatial distribution of CWCs, and there are mixed effect present. In the following paragraph we give an overview of all the different effect studies, and physiological experiments that were conducted with CWC reefs, and of which we can use the data for our model.

We added a sentence that food flux and availability is one of the most important predictors of coral growth, with the suggested reference ##

Lines 113-115: I assume that the strong regional contrasts refer to heterogeneity in geomorphology/terrain. Please rephrase.

Thank you, adapted accordingly. These sentences now read as follows:

The CWC mounds and ridges on the south-eastern (SE) slope of Rockall Bank (northeast Atlantic Ocean) provide an excellent study site to develop a mechanistic model of CWC biomass and respiration in relation to local hydrodynamics and food supply. This area shows strong geomorphic heterogeneity with numerous CWC mounds between 500-1000 m depth which are surrounded by sediments. The CWC mounds in this region have been studied extensively for several decades (e.g., de Froe et al., 2022; van Haren et al., 2014; Kenyon et al., 2003; Mienis et al., 2006). These mounds are formed by the framework building CWC species *Desmophyllum pertusum* (previously known as *Lophelia pertusa*, Addamo et al., 2016) and *Madrepora oculata*, for which a relatively large amount of physiological data is available.

Lines 129: I suggest highlighting that these variables are rarely predicted by statistical models, in the discussion this is an important contribution.

 Thank you, we adapted the sentence to the following:

We specifically aim to predict CWC biomass and quantify CWC and benthic respiration, parameters which are difficult to predict with statistical spatial distribution models.

We emphasized this also in the discussion, see reply on general remarks.

Lines 193-195: I think that an "a" should be added to the reference ( Soetaert et al. 2016a). In the work by Soetaert et al, it is mentioned that POC is influenced by passive sinking, hydrodynamic transport, and biological degradation. The results of this study highlight downwelling events often transferring POC to mounds. In these lines, the authors mention advective and passive transport,

which may lead non-technical readers to think that upwelling/downwelling were not considered. Please clarify this for non-specialists.

The Soetaert et al., 2016a was a reference error. There is only one Soetaert et al. 2016 cited in the paper. We have corrected this now. We changed the text here now to:

A detailed description of the organic matter transport model can be found in Soetaert et al. (2016). In this model, POC transport is simulated in the model domain by a combination of advective transport, passive sinking, and biological degradation. As in Soetaert et al. (2016), advective transport is driven by horizontal currents ($u$,$v$) and vertical or upward/downward currents ($w$).

Line 243: The parameter mCWC does not seem to be standardized to coral size/weight or area. Confirm whether it should be expressed per m$^2$.

Indeed, respiration is here expressed in % biomass respired per day, which is sometimes used in mechanistic models. The CWCs have a base respiration rate of 0,0035 d$^{-1}$, so every day 0,35% of its biomass is respired.

Lines 329-332: Can the authors provide the rationale for these decisions (e.g. dividing initial biomass by 3, coral growth/decline enhancement factor of 12)?

The specific values are chosen iteratively throughout the modelling process. The value of dividing the initial biomass by three was chosen due to:

- The initial biomass of CWC biomass was too overestimated in each grid cell of the model, and dividing by three would still overestimate the CWC biomass in the model domain, but speed up computations considerably.
- If we divided by two, this would also be the case, but computations would just take longer to end up with the same results.
- If we dived by ten, some areas would be underestimated, and so in those areas CWCs would start to grow.
- To synchronize the effect of overestimation throughout the model domain we decided to divide the initial biomass by three.

The coral growth/decline enhancement factor of 12 was chosen due to:

- Corals would grow/decline considerably slower than the changes in POC transport/hydrodynamics.
- We chose 12 for two reasons: 1. As we have one month of hydrodynamic output, a factor 12 would represent one year of growth/decline, which makes it a bit more intuitive to compare speeds. 2. Enhancing the coral growth/decline also affects daily dynamics of corals in the model. Some areas would, for instance, show CWC growth over the whole month period, but would CWCs would decrease for some days/weeks due to lack of POC influxs. Then at spring tide, when POC flux would increase (see video supplement), the CWCs would grow again. By using an enhancement factor, this growth/decline is amplified and if much larger factor is taken (say factor 100), then fluctuations of CWCs would be very high over daily time scales. This also affects then the POC dynamics in the bottom layer, and therefore the

mechanics of the model. A factor of twelve proved to speed up computations, but would not affect the dynamics of the model. This is done iteratively, so other factors were used in earlier versions/in the development of the model.

We have adapted the text to the following to accommodate this comment:

First, we divided the initial CWC biomass from step 2 by three and used that as initial benthic biomass. The specific value of dividing by three was chosen iteratively during the model development process. Dividing by three would speed up computations considerably but would not alter the initial conditions retrieved from step 1 and step 2. The new initial CWC biomass would still be higher than the CWC equilibrium biomass throughout the model domain, as CWC biomass was still declining in consecutive model runs. Second, we ran the coupled model for a total of five consecutive months with a coral growth/decline enhancement factor of twelve. This is a method that is also used in morphological and sediment transport modelling approaches (Ranasinghe et al., 2011). The enhancement factor of twelve was chosen iteratively during the modelling development process and was chosen due to the following reasons: 1) as we used one month of hydrodynamic output, an enhancement factor of twelve would compare with one year of running the model, which makes comparing modelling output with/without enhancement factor more intuitive, and 2) using a much larger factor (i.e., enhancement factor of 100), would change the short-timescale dynamics of CWC growth and thereby alter the mechanics of the model. For example, using an enhancement factor of 100 would in some areas of the model domain cause strong daily CWC biomass fluctuations, which would affect the bottom water POC concentrations as well. A factor of twelve proved to speed up the computations but would not affect the mechanics of the model. After running the model for five consecutive months with the enhancement factor, two months were run without the growth- enhancement to arrive at the final output, in which $dCWC_b / dt$ was close to steady-state.

Line 396: Should this be "local" rather than "regional"? How easily could the model be scaled up to a larger area?

Thank you, I guess it depends on your definition of local or regional, if you have sufficient computer power, you could scale up this model to larger areas. Changed the word to local.

Line 403: "shows" instead of "show."

Thank you, corrected.

Line 412-413: This sentence is a bit confusing. It seems to refer to environmental factors that were included in the model, but that long-term variations in these factors where not considered, potentially leading to discrepancies between observed and actual CWC cover. The reference to respiration may confuse the reader, please rephrase.

Thank you, we see what you mean.

This section discusses the fact that our modelled CWC biomass agrees well with dead coral framework cover, and the causes behind this.  We adjusted the text to avoid confusion, and removed the reference to respiration. The new text now reads as follows:

The good agreement between our modelled CWC biomass and observed combined cover of living CWCs and dead coral framework could have several causes: 1) the different timescales between our model and CWC reef dynamics. Our model is based on one-month of hydrodynamic model output, organic matter transport and CWC physiology. Although CWC biomass declines in the model if insufficient food is provided, mortality or longevity of CWCs is not included. This means that, in our model, if (food) conditions remain favorable, CWCs can exist indefinitely. Although generally little is known on the temporal and spatial dynamics between living CWCs and dead coral framework on a reef, CWCs would die-off at one point in time and become dead coral framework. 2) It could be that CWCs have grown in the past in the areas where we predict high CWC biomass, but which have died-off due to conditions that were not included in our model (i.e., infection, predation, temperature, ocean acidification). Therefore, it is reasonable to find dead coral framework where high CWC biomass is predicted in our model. Previous work on the same video transects also shows that cover of live corals and dead coral framework are highly correlated (van der Kaaden et al., 2023). The presence of dead coral framework on the mounds indicates areas that were favorable for CWC growth in the past. It would be interesting to expand our CWC biomass model with dead coral framework as a state variable, where dead coral framework is built up with a mortality rule (as in Hennige et al. 2021). This would especially be interesting as dead coral framework affects bottom hydrodynamics (Bartzke et al., 2021; Corbera et al., 2022) and baffles sediment (Wang et al., 2021), and alters nutrient cycling (Maier et al., 2021).

Line 237-239: Considering that the simulation was run for a month, how was benthic respiration extrapolated to the whole year? Did the authors assume that POC is steady throughout the year? I also suggest dividing this sentence in two, to improve flow.

We assume that the reviewer refers to lines 437 – 439 instead of 237 – 239.

Yes, in this case we assumed benthic respiration in the model is representative for the whole year. So it would be the integrated respiration of the whole month times twelve. There is no data available on the seasonal fluctuations of CWC benthic respiration. The numbers of de Clippele et al., 2021 are also based on benthic respirtation values measured in spring, so with the same kind of conditions, and extrapolated to the year. So extrapolation was done in a similar matter.

We changed the text to the following:

Extrapolating our modelled benthic respiration to a whole year, the seafloor of the model domain would respire in total 104,845 tonnes C per year. CWCs alone were responsible for 11,260 tonnes C yr$^{-1}$ of benthic respiration, or 10.7% of the total benthic respiration in the model domain.

Line 239: The reference to 5,763-9,260 tones is not standardized to area, was the referenced study at exactly the same domain?

We assume that the reviewer refers to line 439.

Yes indeed, we added this info by changing the sentence to the following:

Our predicted CWC-based respiration was comparable to, but at the upper end of the carbon turnover estimate of 5,763 to 9,260 tonnes C yr$^{-1}$ which was predicted from a CWC suitable habitat model of exact the same area (De Clippele et al., 2021; Rengstorf et al., 2014).

Line 440: Specify whether this reference refers to "a" or "b."

Thank you, done.

Lines 572–575: On a first read, I found the first sentence slightly contradictory to Fig. A and the fact that this study used a POC model which has highlighted the interaction between tidal currents and CWC-formed mounds in the past (Soetaert et al. 2016a). In the second sentence I realized that this part refers to bottom currents and processes that likely occur in smaller scales. I suggest authors to rephrase these two sentences to avoid confusion.

Thank you, indeed confusing. We rephrased the sentence to the following:

The interaction between bottom hydrodynamics and CWC framework was not included in the model as it would require resolving spatial scales much smaller than 250 m (horizontal resolution of one grid cell in this model = 250 m$^2$).

Figures

Figure 1: A: Adding labels on land shapes will help readers that are not familiar with the area. A short note on the relationship between parent grid, child grid and model domain would also help. These are mentioned in the text, but their relationship is a bit unclear.

Thank you, will be adapted accordingly.

Figure 3: Please add latitude and longitude labels.

These will be added##

Figure 7B: Consider comparing modelled versus observed depths directly in panel B for easier interpretation.

This will be added##

Figure 9: For panel A, I suggest using unfilled rectangles to keep the underlying data visible. For panel C, I believe that a simple correlation plot between measured and modelled values (keeping the colour palette for sand and coral areas) would be more informative.

For A): this will be adjusted. ##

For C) we plotted this correlation, but due to the differences in the spatial resolution, the relationship is not 1:1 for each side. See plot below. We think it is more useful to keep comparison on the scale of the whole model domain, rather than a 1:1 regression plot.

[Figure]

Figure 10: Consider overlaying panel E with panels A and C as an inset, rather than showing it separately. Similarly, clarify the locations of panels A and B. Standardizing the format for all figures that refer to the model area (e.g. Figure 1C, 3, 4A,B), with similar contour line style, colour palettes etc. might help the reader follow the figures easier.

Will be adjusted accordingly, thank you for the suggestions. ##

---

## Author Comment (AC2)

**Reply to reviewer 2:**

**General comments**

The paper outlines the development of a new mechanistic model predicting coral biomass and respiration rate at Rockall Bank. Model data is compared to observational data and habitat suitability models from Rockall Bank to assess how the new model performs. As someone who is interested in food delivery to deep-water coral habitats I am very excited about the future possibilities of this type of work. I suggest minor revisions, which are outlines below. If this revisions can be addressed, I would happily support this for publication and look forward to seeing what mechanistic modelling of corals can tell us about nutrient cycling.

**Specific comments**

It would greatly benefit the strength of the paper if the authors could find a way to compare model results to observational data. Maybe an example could involve taking CWC biomass (mmol C/m2) as a ratio of carrying capacity to get a number that is comparable to percentage cover? A few lines of text going over this could replace the section where authors say results are challenging to compare. In doing so, they could elaborate on what type of substrate/conditions/depths the model is most useful for.

The model shows quite good agreement with a published habitat suitability model which is good to see, but more explanation on why the presented model should be chosen over a habitat suitability model moving forward could be explored. An example is given on line 585 where the effect of rising T could be indirectly included in the new model, but presumably T could also be directly updated in a habitat suitability model to predict the effect of warming T? It seems like exploring changes to food availability is a strength of this mechanistic model so expanding on this, or something similar would be very interesting.

Thank you for your review and feedback. To answer your concerns:

1) Comparing model results to observational data.

We see your point. In section 2.5 we elaborate on the data sources that we use to compare our model with observations. We compare our model with coral cover data (the videos, Maier and de Clippele et al.), benthic respiration (the box cores and AEC technique, de Froe et al.), and habitat suitability models (Rengstorf et al.). To use a biomass to carrying capacity ratio could be useful to compare with coral cover, but it is also a bit of a paradox. The carrying capacity was estimated based on pictures from the box cores from de Froe et al., 2019. It is an estimate, and the actual value of this parameter is not well constrained. Coral cover of the video transects is calculated in a different way, as these videos are taken with an ROV. However, from de Froe et al. 2019, we do have biomass estimates from several locations. We will add some sentences that compare our modelled biomass data from these box cores. ##

2) Habitat suitability models vs. mechanistic models: I am not sure if our mechanistic model approach should always be chosen over a habitat suitability model. The two models require different kinds of data as input, which is available/not-available depending on the situation.

Reviewer three also had the following comment on this: Additionally, some key advantages of the approach could be highlighted further. For example, this mechanistic model can estimate parameters that are difficult to obtain with statistical modelling such as species distribution models (e.g., biomass, respiration), especially in data-limited environments such as the deep sea. Statistical models often require extensive datasets and in the case of deep-sea benthic species they are typically limited to presence/absence or abundance data, with very few exceptions, such as the models used for validating the present study. Emphasizing that this model integrates physiological and environmental information without requiring large-scale sampling, except for validation, would strengthen the paper's contribution and practical significance.

To tackle these remarks, we adapted section 3.4 by emphasizing what the advantages are of mechanistic modelling compared to statistical habitat suitability models. For example, in statistical models, temperature could indeed be adjusted as well, but that would be based on a static dataset, where you, for example adjust the mean bottom water temperature over a spatial grid. In mechanistic modelling, you can couple the physiological processes with the water temperature, on a much smaller temporal scale. We emphasize this now in section 3.4. ##

**Scientific questions**

83 – what is meant by interactions?

Thank you, we mean interactions between cold-water corals and their environment. We added this to the sentence.

85 – enhanced – This word choice makes it sound like corals increase the amount of OM. But I assume it's meant to convey that OM concentrations are higher around the reef relative to nearby sediments because of their ability to retain nutrients? Maybe rephrase as "availability is higher on the reefs related to around them"? Or something like this?

Thank you, we adjust this sentence accordingly. ##

166 – Why is suspended POM selected over sinking POM? Suspended POM is usually more refractory than sinking POC? Sinking POM would presumably have a faster settling rate and so could change model outputs. A line or two on the distinction and the decision for the model would be informative.

Thank you. We mean sinking POM, we will update the MS accordingly. ##

- Suspended and sinking POM are the same, nearly all suspended POM also sinks, depending on the size/shape of the particle (Ivensen et al., 2020).
- We chose a sinking rate of 10 m d-1. Choosing a faster settling rate would increa the passive supply of POM to the seafloor but would lower the concentration of POM in the model. Choosing a slower settling rate would decrease passive supply but increase concentration in the model domain.

In the text, could you expand on any environmental factors that could explain why the model has difficulty matching VT6 in figure 8? Such a large presence of corals from 0-250 m coincides

with very low biomass compared to what is predicted in VT2, VT5. Line 386 says the model agrees with observational data but they don't match that well for some sections. What is it about VT2 that leads to such high values? Some specifics on the sites could be useful beyond mentioned model issues (depth of the model and patchy resolution).

Thank you, there are indeed some transects that match less well with our model than others. We mean that in general, the modelled CWC biomass matches high/low coral or dead framework cover. You can see in the transects, except for VT7, that if high coral cover is recorded in the video, our model shows also a peak on that transect. Therefore, we now start the sentence in 386 with "Generally, ".

But there are indeed some differences between the sites. For VT6, the model predictions are shifted somewhat southward, as with VT1. This is due to the general current direction and velocity in the model domain, and the much lower resolution of the model domain compared to the video transects, and the location of the transects on the mound (i.e., northwestern slope or the northeastern slope). As the general current is in southwest direction, most high CWC biomass is found on the south/southwestern slopes of the mounds. For example, VT6 is located on the northeastern slope of Haas mound, and therefore shows a lower biomass in the model, and VT2 is located on the southern slope, and therefore Coral biomass at VT2 is higher than at VT6. We adapted this section so that we clarify where this difference comes from. ##

**Technical revisions**

Line 9 – space between 2 and C which other affiliations don't have.

Done.

Line 22 – The distribution of "these"

Done.

Line 25 –30 It's a bit unclear how many different models are used when reading the abstract. Rephrasing this so it's clearer would help the reader.

Thank you, we changed this to the following sentences:

Here, we present the results of a mechanistic process-based model in which coral biomass and respiration are predicted based on hydrodynamics, organic matter transport and coral physiology. The model domain comprises the cold-water coral mounds of the south-east Rockall Bank in the north-east Atlantic Ocean.

Line 30 – available experimental reports - plural.

Done.

Line 32 – occurrences "comply" – Odd word choice?

Done, replaced with agree.

Line 72 – suggest revising sentence. As stated, this is a long sentence and it might be clearer/more accurate to take the latter half "and new tools and models are needed to understand how CWCs will be influenced by a changing marine environment" and rephrase it to say that new tools and models represent a way to offset or supplement, etc. the challenges of physically sampling.

Line 78 – provided with observation of CWC "habitats"? add word possibly? Or is it supposed to be about the physical traits of corals? As written, it's a bit ambiguous.

Done.

107 – consider adding Girard et al., 2022 - https://doi.org/10.1098/rspb.2022.1033

Very applicable, thank you for pointing that out.

141 – "biogenic soft sediment" < what does the descriptor "soft" add? As if soft rock? (i.e. sedimentary over metamorphic or igneous?). If so, I think this is covered by it being "biogenic". Or soft as in unconsolidated? Consider either removing or replacing word "soft"? If the latter is the intended description, then maybe "unconsolidated biogenic sediment"?

Removed the word soft.

143 – It would be great if you explicitly state the relief of the key mounds above the biogenic sediments.

The end of this paragraph provides a bit more information on the relief. We moved the end of the paragraph to the middle and added information on the height of the mounds and ridges. This paragraph now reads as follows:

The study area is situated on the south-eastern (SE) slope of Rockall Bank (north-east Atlantic Ocean; Figure 1A). The substrate in this area is characterized by biogenic sediment at the shallow part of Rockall Bank (300 – 500 m depth), coral capped carbonate mounds and ridges on the slope between 500 – 1000 m depth, and biogenic sediments in between the carbonate mounds and in the deeper part of the Rockall Bank slope (>1000 m depth; Kenyon et al., 2003; Mienis et al., 2006). Numerous CWC ridges and mounds are found along this slope, in an area known as the Logachev mound province (Figure 1B). The CWC ridges differ in height and shape but are mostly elongated perpendicular to the slope. CWC ridges and mounds are generally between 50 – 300 meter in height. The largest CWC mound in the model domain is called "Haas mound" which is around 300 m high, one to two km wide, five km long, and elongated parallel to the Rockall Bank slope (van der Kaaden et al., 2021). For readability, we will refer to the coral mounds and ridges in this area as 'CWC mounds'.

The current direction throughout the water column is predominantly to the southwest, driven by the clockwise gyre circling the Rockall Bank (Hansen and Østerhus, 2000; Holliday et al., 2000; Mienis et al., 2007; Schulz et al., 2020). The area is subject to internal waves with amplitude of several 100s of meters and high bottom current speeds (i.e., >50 cm s$^{-1}$; Mienis et al., 2007; Mohn et al., 2014). Interaction of tidal currents with mound topography cause breaking of internal waves (Cyr et al., 2016) with subsequent downward transport of organic matter (Duineveld et al., 2007; de Froe et al., 2022; Soetaert et al., 2016).

161 – possible suggested re-write: "The keystone species D. pertusa is our selected model species because it provides habitat/and contribute to is important to reef metabolism.

Thank you, changed the sentence to:

The keystone species *D. pertusum* is used as the model species in this study because it provides habitat to numerous associated organisms (Costello et al., 2005; Freiwald et al., 2002; Husebø et al., 2002; Jensen and Frederiksen, 1992), and contributes substantially to reef metabolism (de Froe et al., 2019).

162 "used as (the?) model species" missing word?

See above.

193 – variably called organic matter transport model, organic matter reactive-transport model, or reactive transport model. Should be consistent throughout.

198 – "/" which should be a "."?

Thank you, corrected.

201 – consider replacing word "representable" with "representative".

Replaced the word.

207 – it would be nice if this formula were in a slightly smaller font and could be on one line.

This should be done in final formatting of the OS editing team I think.

210 – No POCC in formula – is the extra C a type?

Yes, apologies, corrected.

230 – biomass "is" calculated? Typo?

Yes, thank you.

253 – and consists "of" 1.36? typo?

Yes, thank you.

254 – D. pertusum comprises of 2.12 <- should either be "comprises 2.12" or "consists of 2.12".

Corrected.

294 – change in font size

Thank you, adapted.

394 – missing word? Due "to" a patchy distribution

Thank you, corrected.

420 – typo – where framework is "built" up

Thank you, corrected.

426/27 – font change for references

Thank you, corrected.

**Figures**

As a general comment for all map figures, it would be informative to have contour interval either on the map or in the caption. Contour lines are labeled in figure 10C but nowhere else I believe.

Thank you, contour lines will be added##

Figure 1

Missing scale bars, contour labels and geographic labels throughout Figure 1. Figure 1A is not useful if the reader is not familiar with the region. Ideally, UK and Ireland could be labeled. B and C – what are the intervals of the contour lines? Either add to figure or in caption. C) Latitude label is mostly covered and if there was meant to be a longitude label it is missing.

It's also a bit confusing that red arrows convey current trajectory and as a label pointing at the mound. Maybe change the colour of one? Or possibly remove the arrowhead from the Haas mound label?

Thank you, I adapted the figure according to your comments. ##

Figure 2

First word should be capitalised.

Done.

Figure 3

Could you provide a scale quantifying the shade of green used? Presumably darker means higher concentration of POC?

Yes, I will adapt this accordingly. ##

Figure 4

I think coral region and coral presence lines are the same? Could you simplify this to one label so the figure and caption are consistent?

Yes, I will adapt this accordingly. ##

Figure 6

Consider replacing this with panel D from Figure 10. It would be more useful to show how the modeled data compares with the habitat suitability model since that is a focus of the paper.

It is the same data, only in figure 10 is the statistical model shown on top of it as red lines. I suggest keeping it this way, as I think the red lines interfere a bit with our results. So, if I replace this figure with 10D, then the distribution of biomass from our CWC model will be unclear. Now you can see nicely that biomass concentrates on the southern slopes of the CWC mounds.

Figure 7

Why does coral presence reach 40-50% in panel A, but corals are nearly absent in Figure 8F? Shouldn't they both come from video 7?

Thank you, no these are not data from the same video. I see that we say in the caption that this is video 7, but it is actually video 1. I changed this to video 1.

Figure 8

Panel G – should these numbers be 1-7? Not all 1?

The number 1 represent the start of the transect, I replaced it with the "Start", to clarify. ##

---

## Author Comment (AC3)

**Reply to reviewer 1:**

De Froe and co-workers use a predictive mechanistic model to estimate cold- water coral biomass distribution and respiration. Their model successfully reproduces observed reef biomass and respiration patterns. The advantage of this approach can be used to obtain the effect of changing environmental conditions such as ocean temperature, export production or ocean currents.

The study presents a first mechanistic model predicting cold-water coral biomass distribution based on organic matter transport and hydrodynamics. The authors set up the model by cleverly coupling three models, offering a new perspective on the mechanisms driving coral distribution. They demonstrate that coupling organic mater uptake with the cold-water coral model is key to predicting the spatial distribution of these corals.

The authors clearly identify existing gaps in the field and present a study that brings new knowledge and tools that can be applied in future research.

**Scientific quality:** yes, excellent.

**Presentation quality:** The manuscript is clearly written and well-structured. The number of figures, conceptual diagrams, and tables is appropriate, and they are of high quality. The supplemental material is warranted and adds value. The authors also discuss the limitations of their work, and the conclusions are well-supported and justified.

I am very positive towards the study, as the findings are important.

Dear reviewer,

Thank you for having taken the time to review our manuscript and your kind words. Below you can find a reply to each of your specific comments:

**Specific comments and detail points**:

Line 87 : "This feedback between organisms and their environment can greatly affect how they respond to environmental changes: by modifying their own environment, organisms can rearrange their spatial patterns in response to climate change thereby avoiding a tipping point towards extinction (Rietkerk et al., 2021)" could be reworded. The temporal aspects are not fully resolved within the time scale of the study. Also add a reference relevant to corals.

Thank you for this comment. Indeed, the changing spatial patterns are not resolved by our model. This would require a dynamic seabed in our model domain. We here introduce the idea that spatial-self organization can accommodate negative effects of climate change in general (Rietkerk et al.) and that CWC reefs show signs of spatial self-organization (vd Kaaden et al.). We rephrased the sentence to clarify this. ##

References in introduction and discussion : Consider reducing to 3-4, as some currently contain five. This will improve readability and focus.

Thank you. We think it is important to acknowledge research where appropriate. Some sentences indeed have five references, but they are only limited. We suggest leaving these few (extra) references in the manuscript.

Line 163: replace " numerous associated animals" with "numerous associated organisms".

Done.

Lines 508-510: The authors could be more specific in this section to enhance clarity and strengthen the argument.

We interpret this comment as the reviewer would like a more explicit link between depletion of POC in the bottom water, negative scale-dependent feedback, and spatial self-organization. A bit more context is indeed useful to add, we adapted this paragraph as:

Our model results are in line with previous work which shows that CWCs are self-organizing ecosystems engineers and reefs form self-organized regular patterns (van der Kaaden et al., 2023). Self-organization is the process where organisms form regular patterns in ecosystems due to local-scale interactions between organisms and their environment (Camazine et al., 2001). These interactions can either be positive, whereby organisms enhance local resources by modifying their environment (i.e., ecosystem engineers), or negative, whereby organisms deplete resources, which leads to competition (Rietkerk and van de Koppel, 2008). If feedback mechanisms operate at different spatial scales, it is referred to a as scale-dependent feedback, which is a central principle in the theory of self-organization (Rietkerk and van de Koppel, 2008). The results of our modelling study also indicate the presence of scale-dependent feedback in CWC mounds in our study area. CWC mounds enhance food availability on a very local scale (summit and the upper flank of the mounds) which is a positive scale-dependent feedback and leads to high CWC biomass in these areas. Depletion of POC in the bottom water by CWCs in our model decreased the quantity of available food for CWCs on a wider spatial scale located downstream of the areas with high CWC biomass (Figure 10B). This can be seen as a negative scale-dependent feedback (van der Kaaden et al., 2020), and leads to lower CWC biomass in the areas downstream of the upper flank and summits of the CWC mounds. This depletion of resources by CWC in the water column due to filtering activity has also been observed in the field (Lavaleye et al., 2009; Wagner et al., 2011). Our model successfully simulates this process, providing further evidence that CWC reefs are self-organizing ecosystems, and therefore might show resilience to environmental change (van der Kaaden et al., 2023; Rietkerk et al., 2021).

Lines 581 to 585: This section needs to be better explained, particularly the last sentence. I recommend splitting it into two sentences for improved clarity and flow.

Thank you, changed these lines to the following:

The presented mechanistic modelling framework could be used to study the effect of changing temperatures, currents, pH, and nutrients on the spatial distribution of CWCs in the deep sea. CWCs could be severely affected by rising global temperatures and ocean acidification (Chapron et al., 2021; Gómez et al., 2018, 2022; Gori et al., 2016; Guinotte et al., 2006; Lunden et al., 2013; Orr et al., 2005). This modelling framework could be used to study these effects by, for example, coupling CWC respiration to water temperature or mathematically simulating the higher cost of calcification under ocean acidification by increasing basal CWC respiration (Dodds et al., 2007; Gori et al., 2016; McCulloch et al., 2012).

References : There are some inconsistencies in  formatting  within the reference list. Please revise to ensure uniform style throughout.

We revised the reference list and removed any errors (as the reference to Guinotte et al., 2006 for instance).